# An In-Depth Study of Phytopathogenic *Ganoderma*: Pathogenicity, Advanced Detection Techniques, Control Strategies, and Sustainable Management

**DOI:** 10.3390/jof10060414

**Published:** 2024-06-07

**Authors:** Samantha C. Karunarathna, Nimesha M. Patabendige, Wenhua Lu, Suhail Asad, Kalani K. Hapuarachchi

**Affiliations:** 1Center for Yunnan Plateau Biological Resources Protection and Utilization, College of Biological Resource and Food Engineering, Qujing Normal University, Qujing 655011, China; samanthakarunarathna@gmail.com; 2National Institute of Fundamental Studies, Hantane Road, Kandy 20000, Sri Lanka; 3Institute of Chemistry, Chinese Academy of Sciences, Beijing 100190, China; nimesha.patabendige@gmail.com; 4Center of Excellence in Microbial Diversity and Sustainable Utilization, Faculty of Science, Chiang Mai University, Chiang Mai 50200, Thailand; reuven0319@gmail.com; 5School of Biology and Chemistry, Pu’er University, Pu’er 665000, China; sohailasad74@gmail.com; 6College of Biodiversity Conservation, Southwest Forestry University, Kunming 650224, China

**Keywords:** biological control, disease management, pathogens, pathogenicity mechanisms, plant defense responses

## Abstract

Phytopathogenic *Ganoderma* species pose a significant threat to global plant health, resulting in estimated annual economic losses exceeding USD (US Dollars) 68 billion in the agriculture and forestry sectors worldwide. To combat this pervasive menace effectively, a comprehensive understanding of the biology, ecology, and plant infection mechanisms of these pathogens is imperative. This comprehensive review critically examines various aspects of *Ganoderma* spp., including their intricate life cycle, their disease mechanisms, and the multifaceted environmental factors influencing their spread. Recent studies have quantified the economic impact of *Ganoderma* infections, revealing staggering yield losses ranging from 20% to 80% across various crops. In particular, oil palm plantations suffer devastating losses, with an estimated annual reduction in yield exceeding 50 million metric tons. Moreover, this review elucidates the dynamic interactions between *Ganoderma* and host plants, delineating the pathogen’s colonization strategies and its elicitation of intricate plant defense responses. This comprehensive analysis underscores the imperative for adopting an integrated approach to *Ganoderma* disease management. By synergistically harnessing cultural practices, biological control, and chemical treatments and by deploying resistant plant varieties, substantial strides can be made in mitigating *Ganoderma* infestations. Furthermore, a collaborative effort involving scientists, breeders, and growers is paramount in the development and implementation of sustainable strategies against this pernicious plant pathogen. Through rigorous scientific inquiry and evidence-based practices, we can strive towards safeguarding global plant health and mitigating the dire economic consequences inflicted by *Ganoderma* infections.

## 1. Introduction

The presence of pathogenic *Ganoderma* species across the globe poses a significant danger to several plant species, resulting in a considerable economic impact on the agricultural and forestry sectors [1,2]. These fungal pathogens belong to the genus *Ganoderma* (Figure 1) and are known to cause severe diseases in a wide range of plant hosts [3,4]. The genetic diversity and phylogenetic structure of *Ganoderma* reveal complex relationships between different species, reflecting their evolutionary history and adaptation strategies. Studies using molecular tools such as DNA sequencing have helped elucidate these relationships, highlighting the significant variability among species that influences their pathogenicity and ecological roles [5]. This genetic insight is crucial for accurately identifying species and understanding their interactions with host plants, which is fundamental for managing the diseases that they cause. *Ganoderma* comprises several species, including *Ganoderma boninense*, *Ganoderma lucidum*, and *Ganoderma applanatum*, among others, each capable of causing devastating diseases in plants [6,7,8,9]. Phytopathogenic *Ganoderma* infections typically begin through root contact, wounds, or natural openings, such as stomata. Once established, the pathogen colonizes the host tissues, leading to the development of symptoms such as wilting, yellowing, decay, and, ultimately, the death of the plant [10,11]. The life cycle of *Ganoderma* involves the production of spores, which are dispersed through various means, including wind, water, and human activities, contributing to the widespread dissemination of the pathogen [12,13].

The pathogenicity mechanisms employed by *Ganoderma* species involve the secretion of various enzymes and toxins that facilitate the invasion and destruction of host tissues, including ligninases, cellulases, proteases, laccases, phytotoxins, and secondary metabolites such as ganoderic acids [14,15]. Furthermore, environmental factors, such as temperature, humidity, and soil characteristics, are critical in *Ganoderma* infection and disease development [16,17]. Understanding these factors’ influence is essential for designing and implementing effective disease management strategies. In addition to pathogen biology and ecology, the intricate interactions between *Ganoderma* and host plants are vital for disease management [18,19]. *Ganoderma* employs colonization strategies to establish itself within the host, overcoming plant defense mechanisms [20,21]. The pathogen also triggers a series of defense responses in the plant, and understanding these interactions can help develop strategies to enhance plant resistance and suppress *Ganoderma* proliferation [22,23].

Implementing integrated disease management approaches is crucial for effectively controlling phytopathogenic *Ganoderma* [24,25]. Cultural practices, such as the use of disease-free planting materials, proper sanitation, and crop rotation, can help prevent the introduction and spread of the *Ganoderma* pathogen [18,26]. Chemical control methods, including the use of fungicides, can provide some control, but their long-term sustainability and environmental impact need careful consideration [27,28]. Biological control agents have emerged as promising alternatives for *Ganoderma* management [28,29]. Antagonistic microorganisms, such as bacteria, fungi, and actinomycetes, have shown potential in suppressing *Ganoderma* growth and disease development [25,30]. Biocontrol agents, such as *Trichoderma* spp., *Bacillus* spp., and *Streptomyces* spp., have been found to stimulate plant growth and induce systemic resistance against *Ganoderma* [31,32,33].

Another critical aspect of *Ganoderma* disease management is plant resistance breeding. Identifying and utilizing resistant germplasm, studying resistance mechanisms, and developing molecular markers for marker-assisted selection are important components of breeding programs aimed at developing resistant cultivars [34,35,36]. Hence, understanding the biology and ecology of pathogenic *Ganoderma* species is crucial for implementing effective control measures and managing the diseases that they cause. This review provides a comprehensive overview of phytopathogenic *Ganoderma*, focusing on control measures and disease management strategies.

## 2. Ecological Dimensions of *Ganoderma*

*Ganoderma* play significant ecological roles in forest ecosystems. They are wood-decaying fungi found on dead or dying trees, fallen logs, and decaying wood debris [37]. As decomposers, they contribute to the process of wood decay, playing a crucial role in nutrient cycling and carbon sequestration [38]. *Ganoderma* species break down complex organic compounds, releasing nutrients into the ecosystem and facilitating the recycling of vital elements [39]. Additionally, *Ganoderma* form mycorrhizal associations with host trees, forming a mutually beneficial relationship. They colonize the roots of trees and form mycelial networks, enhancing nutrient uptake, particularly phosphorus, and water absorption by the trees [40]. In return, trees provide carbohydrates to the fungi, supporting their growth and reproduction. This mutualism contributes to the overall health and resilience of forest ecosystems [41,42]. *Ganoderma* also interact with various organisms in their environment. Recent studies have highlighted their role as a habitat and food source for diverse invertebrates, such as beetles and springtails [43,44]. These interactions influence the abundance and diversity of associated organisms, contributing to the overall biodiversity of forest ecosystems [45]. Overall, the ecological aspects of *Ganoderma* encompass their roles as decomposers, as mycorrhizal partners with trees, and in interactions with other organisms in forest ecosystems [46,47]. Understanding these ecological functions is crucial for assessing the ecological impacts of *Ganoderma* and their conservation in natural habitats.

As previously described, *Ganoderma* spp. are important plant pathogens, causing a significant impact on various crop and forest tree species [48]. One of the most notorious examples is *G. boninense*, which causes the devastating basal stem rot disease in oil palm plantations [49,50]. This pathogen infects the lower stem of oil palm trees, leading to the decay of the vascular tissues and eventual tree death [51]. In addition to oil palm, *Ganoderma* species have also been reported as pathogens in other economically important crops, causing severe diseases such as stem, butt, and root rot, especially in tropical countries [52,53]. Furthermore, *G. lucidum* has been associated with root rot disease in cultivated mushrooms, such as *Agaricus bisporus*, and it affects the root systems of mushroom crops, causing reduced yield and quality [54]. Moreover, *Ganoderma* species can also cause diseases in forest tree species [55,56]. *Ganoderma applanatum* has been identified as a pathogen causing white-rot disease in various tree species, including oaks and pines. This pathogen colonizes the heartwood of infected trees, leading to the degradation of lignin and cellulose, ultimately causing wood decay [57]. Recent studies have focused on understanding the pathogenicity mechanisms of and developing management strategies for *Ganoderma*-related diseases [37,58,59]. These findings can contribute to developing diagnostic tools and targeted management approaches for basal stem rot in oil palm. Furthermore, *Ganoderma* species have the potential to play a vital role in horticulture by infecting landscape plants, including *Acacia* sp., *Cassia* sp., *Pinus* sp., and fruit trees like Avocado [60]. A comprehensive list of the pathogenic, wood-decaying, and/or wood-inhabiting members of Ganodermataceae, the diseases that they cause, and the corresponding host plants is listed in [6].

## 3. *Ganoderma* Life Cycle

The life cycle of *Ganoderma* involves several stages, from spore germination to the production of mature fruiting bodies (Figure 2). The life cycle begins with the germination of *Ganoderma* spores. Spores are released from mature fruiting bodies and can be dispersed through various means, including wind, water, and human activities. Germination occurs when favorable conditions, such as moisture and suitable substrates, are present [12]. Once the spores germinate, they develop into hyphae, which are thread-like structures. Hyphal growth occurs as the hyphae extend and penetrate into the surrounding substrates, such as wood or plant tissues, for nutrient acquisition [12,13]. As the hyphae grow, they colonize and invade the host tissues. *Ganoderma* species employ various strategies to overcome plant defenses and establish themselves within the host. This colonization leads to the development of symptoms and disease progression in the infected plants [20,21]. Under favorable conditions, *Ganoderma* species can develop fruiting bodies, also known as basidiocarps. Fruiting body formation is a complex process involving the differentiation and maturation of specialized structures for spore production. The mature fruiting bodies release spores, continuing the life cycle [12,13]. It is important to note that the specific details of the *Ganoderma* life cycle may vary among different species and environmental conditions.

## 4. *Ganoderma* Basal Stem Rot Disease in Oil Palm

*Ganoderma* basal stem rot (BSR) is a devastating disease that affects oil palm (*Elaeis guineensis*) plantations worldwide. It is caused by the fungal pathogen *G. boninense*, which infects the basal portion of the oil palm stem, leading to significant yield losses and even plant death [2]. Yield losses can reach up to 80% in severely affected plantations, making BSR one of the most economically damaging diseases in the oil palm industry [17]. BSR is considered one of the major constraints in the oil palm industry, posing a significant threat to oil palm production and profitability. The disease spreads through the soil and primarily infects older oil palm trees, although younger trees can also be affected. Infected trees show symptoms such as wilting, frond drooping, and the development of a white rot at the base of the stem. As the disease progresses, the affected area becomes soft and spongy, leading to the collapse of the palm [17]. BSR management strategies typically involve a combination of cultural practices, agronomic measures, and chemical control methods. These include regular field sanitation, the removal and destruction of infected palms, soil mounding, and the use of fungicides [2]. However, complete eradication of the disease has proven challenging due to the persistence of the pathogen in the soil and the lack of complete resistance in oil palm cultivars [61].

Efforts have been made to develop resistant oil palm varieties through breeding programs, as partial resistance has been observed in certain oil palm genotypes [62]. Additionally, research has focused on early detection methods, such as molecular and remote sensing technologies, to identify and manage BSR-infected palms more effectively [2]. Hence, *Ganoderma* BSR poses a significant threat to oil palm cultivation, causing substantial economic losses. The continued research and development of integrated management strategies, including the development of resistant varieties and early detection methods, are crucial for mitigating the impact of this devastating disease on oil palm plantations. The following section presents a comprehensive overview of BSR in oil palm. It includes a discussion on several key aspects of the disease, including its geographical distribution, symptomatology, economic importance, and epidemiology, which involves examining the roles of mycelium contact and *Ganoderma* basidiospores. Additionally, it explores the predisposing factors associated with BSR, such as the age of the oil palm, previous crops, soil type, nutrient status, and planting techniques. This review also delves into early detection methods and outlines various management strategies for BSR, encompassing cultural, chemical, and biological control approaches.

## 5. Plant Defense Mechanisms of BSR

Plant defense mechanisms are crucial in protecting against BSR, a devastating disease caused by *G. boninense* in oil palm. Recent studies have provided insights into the defense strategies employed by plants against BSR (Figure 3). For instance, the expression of the polygalacturonase-inhibiting protein (*PGIP*) gene has been identified as a potential biomarker for disease resistance in oil palm [63]. Additionally, the differential expression of genes such as *SAD1*, *SAD2*, *MT3-A*, and *MT3-B* has been observed in oil palm seedlings during *G. boninense* infection, suggesting their involvement in the defense response [64]. These findings highlight the importance of understanding plant defense mechanisms to develop effective strategies for managing BSR in oil palm plantations. However, the next section discusses the research work on the plant defense mechanisms of BSR.

The expression of defense-related genes in oil palm roots infected with *G. boninense* has been investigated, and the findings highlight the influence of salicylic acid (SA) and jasmonic acid (JA) pretreatments on gene regulation during *G. boninense* infection. This research provides valuable insights into the role of phytohormones in the defense mechanisms of oil palm against *Ganoderma* [22]. BSR disease in oil palm is affected by lignin content and salicylic acid (SA) biosynthesis. Recent research has uncovered the physiological, biochemical, and molecular defense mechanisms involved in early BSR symptoms. A gene expression analysis has identified markers associated with defense responses and biosynthesis signal transduction compounds, including the potential use of the *EgLCC24* gene as a molecular marker for *G. boninense* resistance in oil palm seedlings. This study enhances our understanding of oil palm’s defense mechanisms against BSR [65].

Understanding the gene expression patterns in oil palms infected with *G. boninense*, the causal agent of basal stem rot, reveals the activation of salicylic acid and jasmonic acid defense pathways during different infection stages. These defense mechanisms can partially hinder the progression of basal stem rot, but complete eradication of the *Ganoderma* infection remains challenging. A temporal gene expression analysis provides valuable insights for developing strategies to mitigate the impact of this disease on oil palm plantations [66].

Another study described the development of a unique polyethylene glycol (PEG)-mediated protoplast transformation system for *G. boninense*. By optimizing transformation parameters, the study achieved improved efficiency, resulting in a significant number of putative transformants. A molecular analysis confirmed successful gene integration and expression in the transformed *G. boninense*. Establishing this transformation system opens up new opportunities for studying gene functions specifically related to the pathogenesis of *G. boninense* in oil palm [67].

BSR disease in oil palm is influenced by lignin content and SA biosynthesis. Recent research uncovered physiological, biochemical, and molecular defense mechanisms against BSR. A gene expression analysis identified defense markers, including the potential use of *EgLCC24* as a molecular marker for *G. boninense* resistance. These insights advance our understanding of oil palm’s defense against BSR, aiding disease management strategies [68]. Metabolomics profiling using GC-MS and PCA revealed distinct metabolite variations in oil palm roots infected with *G. boninense*. The analysis identified several compounds, including steroidal compounds and fatty acid derivatives, which were more abundant in infected roots than in non-infected ones. These metabolites are known to be involved in pathogen defense and regulate responses to *G. boninense*. The findings suggest their potential role in the defense mechanism against this pathogen [69].

Different oil palm genotypes exhibit varying responses to this pathogen. Tolerant genotypes have a lower lignin content and higher S/G ratios, indicating potential differences in defense mechanisms. Elemental nutrient deposition in the roots also varies among genotypes, with susceptible ones showing a higher content. These observations contribute to our understanding of the host–pathogen interaction in the basal stem rot of oil palms [70,71].

This pathogenic fungus possesses a range of carbohydrate-active enzymes (CAZymes) and cell wall-degrading enzymes (CWDEs), similar to closely related fungi. These enzymes are involved in cell wall degradation and plant–host interactions, contributing to the infection process. Understanding the role of these enzymes enhances our knowledge of *G. boninense* and its interaction with oil palms [14]. *Ganoderma boninense* produces a necrosis-inducing protein (NEP) that triggers necrosis in tobacco and tomato plants. The NEP requires calcium ions for its action and is regulated by salicylic acid and jasmonic acid. This discovery enhances our understanding of NEP’s role in disease development [72].

Polyisoprenoid expression in oil palm tolerant to *G. boninense* was analyzed using 2D-TLC. A diverse expression of polyisoprenoids was observed, indicating a biochemical response to the fungus. Specific polyisoprenoids were detected in leaf and infected root tissues, suggesting their potential as markers for screening oil palm tolerance to *G. boninense* [73]. Oil palm’s susceptibility to *Ganoderma* infection and basal stem rot (BSR) was explored. A gene expression analysis revealed the upregulation of defense-related genes, such as chitinases and glucanases, and the suppression of photosynthesis-related genes in infected leaves. These findings contribute to understanding oil palm’s defense mechanisms and identifying markers for *Ganoderma* detection [74].

The early molecular defense responses of oil palm during its interaction with *G. boninense* were investigated in this study. A gene expression analysis revealed the activation of defense-related genes, a reduction in fungal presence, and the upregulation of reactive oxygen species (ROS)-related genes. Distinct defense phases were observed, with a shift from biotrophy to necrotrophy. These findings contribute to our understanding of the dynamic defense process and the potential use of biomarkers for the early detection of the oil palm–*G. boninense* interaction [75].

By investigating the defense mechanisms of oil palm against *Ganoderma* infection, the study observed increased activity of enzymes and PR proteins as a response to the pathogen. These responses occurred prior to visible disease symptoms. Different nutrient levels were found to influence secondary metabolites, such as phenolic compounds, flavonoids, terpenoids, and alkaloids involved in plant defense. The findings indicate the potential use of these compounds as early detection markers for basal stem rot [76].

Oil palm ecosystems face threats from *G. boninense*, and BSR spreads through mycelia invasion and airborne basidiospores, leading to cell wall breakdown and reduced yield. Plants produce secondary metabolites for defense, but more research is needed. This emphasizes the importance of secondary metabolites in plant defense and explores fertilizer potential in suppressing BSR [77].

*Ganoderma boninense* was studied to understand the role of cyclophilin (CYP) in its pathogenicity. Five CYP transcripts were isolated, showing variations in coding and untranslated regions. Their expression suggests involvement in fruiting body development, stress response, and pathogenicity [78]. *Ganoderma* infection alters lignin and phenylpropanoid pathways in oil palm roots, impacting disease severity, seedling growth, and enzyme activities. Changes in lignin content and composition occur during infection, with an initial increase followed by a decrease. The study enhances our understanding of these pathways during *Ganoderma* infection in oil palm roots [79]. A comparative proteomic analysis of *G. boninense* and *G. tornatum* revealed differences in protein expression related to virulence, pathogenicity, growth, metabolism, and stress. These findings enhance our understanding of the interaction between *Ganoderma* species and oil palm [80].

Optimized protein extraction protocols are important for studying plant–fungus interactions. This study found that the phenol/ammonium acetate in a methanol protocol was the most effective, resulting in reproducible gels and a high protein concentration. This method enabled the identification of ten protein spots and is suitable for studying *Ganoderma* fungi using proteomic approaches [81]. A proteomic analysis revealed species-specific differences in 24 proteins involved in photosynthesis, signaling, stress/defense, and energy metabolism during the *Ganoderma* infection of oil palm leaves. These findings enhance our understanding of the interaction between oil palm and *Ganoderma* species [82]. Oil palm seedlings respond to *G. boninense* infection by regulating genes involved in phenylpropanoid and flavonoid pathways, as well as cell wall formation. Gene expression changes indicate dynamic defense mechanisms and disease resistance against the fungus [83]. The molecular responses of oil palm to *G. boninense* were investigated to enhance disease control, and antagonistic effects of jasmonate and salicylate hormones on defense responses were observed. *G. boninense* regulates ethylene biosynthesis and reactive oxygen species, while oil palm strengthens cell walls and produces antifungal compounds. *Trichoderma harzianum* improves nutrient status. These findings contribute to strategies for preventing and treating basal stem rot [84].

Oil palm progenies were evaluated for resistance to *G. boninense*. A VOC analysis using GC-MS identified eight major phytocompounds, including fatty acids and phenols. These compounds have potential roles in plant defense against stress. They may serve as biomarkers for evaluating disease resistance and aid in developing resistant oil palm varieties [85]. Thiamine pyrophosphate (TPP) and its role as an enzymatic cofactor were investigated in relation to plant stress. The gene transcripts of *THI1*/*THI4* and *THIC*, involved in thiamine biosynthesis, were analyzed in oil palm infected with *G. boninense*. An increased expression of *THI1*/*THI4* and a slight increase in *THIC* were observed in infected plants, suggesting a potential protective role of thiamine against stress during infection [86].

The phospholipid-transporting ATPase (*pde*) gene is being studied in the molecular interaction between *Ganoderma* and oil palm. A cDNA fragment of the *pde* gene from *G. boninense* was amplified using degenerate primers. A sequence analysis revealed a similarity of about 80% to pde genes found in other plant pathogenic fungi. These insights enhance our understanding of the genetic aspects of the *Ganoderma*–oil palm interaction [87]. Three cDNA sequences encoding chitinase-related proteins were cloned from oil palm. The expression of these genes was analyzed in oil palm seedlings treated with *G. boninense* or *T. harzianum*. An increased expression of *EgChit3-1* was observed in the roots of the treated seedlings, while *EgChit1-1* and *EgChit5-1* showed varied expression patterns. A higher expression of *EgChit1-1* and *EgChit3-1* was observed in the leaves of the treated seedlings [88].

Oil palm’s defense response to *G. boninense* infection was examined, revealing an increased expression of defense-related genes in infected roots, and two potential biomarkers for the early detection of basal stem rot were identified by comparing gene expression in *G. boninense* and *Trichoderma harzianum*-infected leaves. This offers promising tools for early disease detection [89]. A proteomics analysis revealed differentially expressed proteins in oil palm roots infected with *G. boninense*. Glucanase and other defense-related proteins were identified, suggesting their role in the plant’s defense response. These findings contribute to understanding the molecular aspects of oil palm’s defense against *G. boninense* and provide potential targets for disease management [90].

The transcriptional responses of oil palm roots infected with *G. boninense* were analyzed. Differentially expressed genes involved in defense mechanisms were identified. These findings have implications for understanding oil palm defense and breeding tolerant varieties [91]. Chitinase expression in oil palm leaves infected with *G. boninense* and treated with *T. harzianum* was studied. *EgCHI1* showed significant upregulation in leaves treated with both fungi, indicating its potential role in defense against basal stem rot disease. However, chitinase expression was not prominent in leaves treated with *G. boninense* alone. This study highlights the complex interplay between chitinases, pathogen infection, and biocontrol treatment in oil palm defense mechanisms [92].

Basidiospores play a crucial role in initiating basal stem rot disease in oil palm. A study found that only dikaryotic mycelia, not monokaryotic mycelia, were able to cause infection. The severity of disease symptoms was positively correlated with the size of the inoculum, indicating that a larger inoculum led to more severe symptoms. These findings highlight the indirect contribution of basidiospores to basal stem rot infection by colonizing available substrates in oil palm fields [93]. The pathogenicity of this fungus on coconut seedlings was investigated using various diagnostic methods. Symptoms and immunological assays confirmed the presence of the pathogen, and PCR provided accurate identification. Histopathological studies supported the pathogenicity. These methods are valuable for screening resistance in planting materials [94].

*Ganoderma boninense*, isolated from oil palm, was tested for its pathogenicity by inoculating seedlings with wood blocks from a rubber tree. All seedlings in contact with the inoculum blocks became infected, displaying progressive leaf desiccation and the development of bracket-shaped sporophores. Longitudinal sections revealed decay in the bole and stem, with hyphae detected in various root tissues. Gummosis and tyloses in xylem vessels were also observed [95].

## 6. Research on Detection Methods for Basal Stem Rot

Oil palm is a valuable crop but is vulnerable to *Ganoderma* infection, which lowers its productivity. Traditional ground-based disease detection methods are time-consuming and expensive [96]. Detection methods for BSR aim to identify the presence of *G. boninense* in oil palm trees, allowing for timely disease management and the prevention of its spread [61,97]. As early detection is crucial in controlling *Ganoderma* disease in oil palms, several approaches have been employed to detect BSR: visual inspection, molecular techniques, immunological assays, chemical composition analyses, geospatial technology, remote sensing with advanced techniques such as multispectral and hyperspectral data analyses, machine learning, electrical properties assessments, terrestrial laser scanning (TLS), spatial mapping, intelligent electronic noses (e-noses), image processing, ANNs, CNNs, GANs, tomography techniques, and RADAR sensors [98,99,100,101]. However, current diagnostic methods lack a direct relationship with pathogenesis, necessitating the development of highly sensitive techniques [102]. *Ganoderma boninense* aggressiveness varies, impacting disease transmission and host damage. Disease severity, vegetative growth, and molecular methods were used to differentiate aggressiveness, and combined methods enabled accurate detection and identification [1]. The following part gives more details on the detection methods for BSR, and it will serve as a valuable resource for researchers interested in monitoring the development of the disease, providing insights into important techniques and methods for detection.

### 6.1. Biomarker Analysis

The biomarkers involved in the early interactions between oil palm and *G. boninense* were identified. Specific compounds in an oil palm extract showed defense potential, while *G. boninense* produced infection-facilitating compounds. Changes in compound concentrations indicated early infections, suggesting biomarkers for non-invasive *Ganoderma* detection in oil palm [21]. A 1H NMR-based metabolomics analysis revealed 20 biomarkers for basal stem rot (BSR) in oil palm. These biomarkers differentiate disease severity and demonstrate potential for early detection and better management in the industry [103]. A mating-type biomarker was developed to detect pathogenic *Ganoderma* species, including *G. boninense*, in oil palm. It offers effective screening for basal stem rot disease and provides insights into disease pathogenesis and management [104].

Improved detection and quantification of *Ganoderma*-induced BSR in oil palms were achieved using ergosterol detection and microwave-assisted extraction (MAE). A positive relationship between ergosterol content and infection duration was found. Thin-layer chromatography (TLC) shows potential for field surveys [105]. A lower chlorophyll content is observed in oil palm seedlings with higher *Ganoderma* disease severity, indicating a negative correlation. Chlorophyll measurement using a SPAD meter is a more accurate indicator of early *Ganoderma* infection than the disease severity index (DSI) [106]. A GC-MS analysis of oil palm trunk tissues identified potential biomarkers for *Ganoderma* infection. Compounds like dodecanoic acid methyl ester and methyl stearate showed promise in detecting the disease [107]. Biomarkers for the early detection of *G. boninense* infection in oil palm were identified using lipidomic profiling. Significant metabolites were discovered, providing potential for early detection methods [108].

A metabolomics analysis identified potential biomarkers for *G. boninense* infection in oil palm. Tolerant palms showed high levels of mannose, xylose, glucopyranose, myo-inositol, and hexadecanoic acid, while susceptible palms had elevated cadaverine and turanose levels. Metabolomics offers insights for screening resistant oil palm varieties against *G. boninense* [109]. Researchers identified the most stable reference genes for gene expression studies in oil palm with basal stem rot (BSR) caused by *G. boninense*. These findings improve data normalization and enhance our understanding of oil palm’s response to BSR disease [110]. Molecular markers for BSR disease in oil palm were identified. Progeny type KA4G1 showed tolerance to BSR, and markers associated with the disease were found. These findings can contribute to the development of BSR-resistant oil palm varieties [111]. The potential of detecting *Ganoderma* ergosterol as a reliable method for the early detection of BSR was explored in this study. The researchers found a strong correlation between ergosterol levels and disease severity, indicating its potential use for quantifying BSR progression and early detection [112]. Ergosterol, a key fungal membrane component, was isolated and characterized from *G. boninense* mycelium. Its structure was confirmed using various analytical techniques. Ergosterol shows promise as a biomarker for quantifying fungal biomass due to its absence in plants and other microbial cells [113]. A method was implemented with the use of volatile compounds as biomarkers for BSR-infected trees. Molecularly Imprinted Polymer (MIP) sensors based on these biomarkers formed an Application-Specific Electronic Nose (ASEN). Successful detection and identification of volatile compound biomarkers for BSR disease in infected oil palm trees have been demonstrated [114].

### 6.2. Chemical Composition Analysis

GC-MS and PCA were used to analyze the metabolite profiles of healthy and *G. boninense*-infected oil palm rachis extracts. Higher levels of choline phosphate and 2-oxoglutaramate were observed in infected samples. This demonstrates the potential of these techniques for assessing *Ganoderma* infection in oil palm [101]. *Ganoderma boninense* infection induces significant metabolic changes in oil palm leaves, as observed through 1H NMR metabolomic profiling. Infected leaves display elevated levels of choline, asparagine, and gallic acid, while non-infected leaves show higher levels of sucrose and indole-3-acetic acid. These findings shed light on the metabolic impact of *G. boninense* on oil palm leaves [103]. A GC-MS analysis of *Ganoderma*-infected oil palm wood tissue detected VOCs using the HS-SPME method. The optimized method was highly reproducible and revealed diverse VOC profiles. Abundant VOCs included 1-octen-3-ol and 3-octanone, potential biomarkers for BSR disease detection, pending further validation [115].

BSR disease in oil palm was detected using microwave-assisted extraction (MAE) and thin-layer chromatography (TLC). Ergosterol levels in infected roots correlated with disease severity. A TLC analysis proved reliable and cost-effective, offering efficient BSR detection in oil palm [116]. A study assessed oil palm progenies for *Ganoderma* BSR tolerance using lignin content as a screening trait. However, lignin accumulation did not correlate with BSR susceptibility or tolerance, indicating its unreliability for screening purposes [117]. A simplified and efficient method for detecting basal stem rot (BSR) caused by *Ganoderma* in oil palms was developed. The method involves microwave extraction and TLC analysis of ergosterol, allowing for a rapid on-site analysis and prompt disease control measures [118]. Novel methods for detecting *G. boninense* infection in oil palm were investigated. Syringic acid (SA) exhibits antifungal properties and decreases in infected tissues. Chitosan promotes SA accumulation, and SA inhibits *G. boninense* growth, indicating potential for developing resistant oil palm cultivars [119]. Ergosterol levels correlate with *G. boninense* infection in oil palm. It can be used for the early detection and effective management of BSR. Further research is needed for its application in BSR control [120].

### 6.3. Electrical Properties Assessment

A modified carbon electrode with rGO and ZnO-NPs detects stress in oil palm leaves caused by *G. boninense* infection. It correlates well with stressed leaf extracts at different time points. The limit of detection for stress was determined at 14- and 30-days post-infection [121]. Metabolomics and an electrochemical sensor with multiwalled carbon nanotubes detected *Ganoderma* infection in oil palm leaves, demonstrating potential for early detection and improved performance with additional components [122]. Dielectric constant (DC) effectively detects the severity levels of *Ganoderma*-induced basal stem rot (BSR) in oil palm trees, while chlorophyll content is less useful. DC can be used as a sensing parameter for BSR disease detection in oil palms [123]. Oil palm leaves’ dielectric properties enable the early detection of BSR. An impedance analysis shows high accuracy in classifying severity levels. Dielectric spectroscopy shows promise as an early BSR detection method in oil palm [124,125]. The electrical resistance (ER) method shows promise for detecting *G. boninense* infection in oil palm trees. Infected trees display lower ER readings, while healthy trees exhibit higher readings. ER could be an effective tool for identifying oil palm trees affected by BSR disease caused by *G. boninense* [126].

### 6.4. Field Studies including Data Analysis

The transmission of BSR from infected oil palm stumps to young seedlings was observed, with low seedling mortality but the presence of *G. boninense* in seemingly healthy seedlings. Genetic diversity indicated the spread of basidiospores and dikaryons, complicating BSR control in oil palm plantations [127]. Indigenous soil microorganisms are vital in suppressing *G. boninense* growth in oil palm. Sterilized soil without these microorganisms led to increased disease severity and reduced seedling growth. Maintaining a healthy soil microbial community is essential for effective disease management in oil palm plantations [128]. The peatland cultivation of oil palms leads to nutrient deficiencies, including copper and zinc. Lower levels of these nutrients are associated with higher *Ganoderma* occurrence in infected areas. This highlights the vulnerability of oil palms on peatlands to *Ganoderma* and nutrient deficiencies [129]. The distribution of *Ganoderma* species in oil palm plantations in Malaysia was studied; *G. zonatum* was the most prevalent species, followed by *G. boninense* and *G. miniatocinctum*. Upper stem rot (USR) was more significant, with *G. zonatum* exhibiting aggressive and random distribution patterns. Additionally, higher infestations of *Ganoderma* were linked to deficiencies in Cu and Zn [130]. BSR was observed in Desa Bukit Kijang’s oil palm plantation in the Asahan region, Indonesia, with varying incidences across different locations. Higher disease occurrence was associated with a higher sand content in the soil [131]. In Cameroon, 17 *Ganoderma* species were identified, including 2 new records and 4 previously unreported species. Oil palm exhibited the highest species diversity among the host plants. These findings contribute to the checklist of 23 *Ganoderma* species in southwestern Cameroon, enhancing the knowledge of *Ganoderma* diversity in the region. A nursery screening method has been developed to assess *Ganoderma* resistance in oil palm progenies. This method involves exposing germinated oil palm seeds to infected rubber wood blocks, enabling the testing of hundreds of crosses per month. Over 1000 crosses have been evaluated, showing a strong correlation with field results and aiding in selecting resistant progenies for commercial planting [132]. The impact of *G. boninense* on oil palm has been explored in a field trial. Proximity to the inoculum source affects disease occurrence, with closer seedlings being more susceptible. Infection occurs when trunks are mounded with soil, while seedlings around uncovered trunks remain symptom-free. A molecular analysis of collected isolates is underway to investigate pathogen spread [133].

Thirteen color indices were analyzed in a study to distinguish healthy and BSR-infected oil palm trees. The G index from frond number 9 proved to be the most effective, with a high correlation coefficient (R = −0.962) and significant separation between the two categories. The power and S models utilizing the G index achieved the highest R2 value of 0.985 [134]. The effects of different *G. boninense* isolates on oil palm seedlings were evaluated. The aggressive isolate caused severe disease and reduced growth, especially in younger seedlings. Consideration of isolate aggressiveness and seedling age is crucial for effective *Ganoderma* control [135].

### 6.5. Visual Inspection

BSR caused by *Ganoderma* is prevalent in oil palm plantations in Southwestern Cameroon. Surveys revealed BSR symptoms and varying incidence rates (5.4% to 39.0%) across estates. Fine sand content positively correlated with disease, while the C/N ratio negatively correlated with disease [136]. Surveys were conducted in Thailand to assess BSR in oil palm plantations. No symptoms of BSR were observed, except for sporophore development in one location. Ongoing monitoring will provide valuable data for disease management and early detection [137]. *Ganoderma boninense* isolates in oil palm seedlings were differentiated based on disease severity and growth measurements. Isolate 5B showed the highest aggressiveness, leading to seedling mortality. Accurate disease detection was achieved using combined conventional and molecular methods [1].

### 6.6. Intelligent Electronic Nose (E-Nose)

An e-nose detects *Ganoderma* infection early and distinguishes species and infection levels accurately using an SVM. High accuracy was achieved, with some overlaps in root samples [138]. An electrochemical sensor with functionalized MWCNTs detects *G. boninense*-related secondary metabolites in oil palms, enabling fast and reliable disease identification [139]. Electronic nose technology is being explored to detect *G. boninense* in oil palm trees. A study identified five VOC markers, including one specific to *G. boninense*-infected palms. Further optimization and field testing are needed to utilize these findings effectively [140]. An Application-Specific Electronic Nose (ASEN) has been proposed for detecting *G. boninense*. The ASEN utilizes specific sensor arrays and Artificial Neural Network (ANN) classification models for effective detection. Initial results show promising potential for this low-cost and non-destructive method in disease detection and plant disease monitoring [141].

Electronic noses are intelligent instruments used for odor classification in various industries. The focus is designing a portable e-nose, sensor selection, and data processing techniques. The e-nose shows potential in detecting *G. boninense* based on odor profiles [142]. A cost-effective e-nose has been developed for detecting *G. boninense* in palm oil tree trunks. It utilizes metal oxide sensors and an Artificial Neural Network algorithm for odor recognition, enabling the rapid response detection of *G. boninense* within minutes. This advancement in plant disease detection technology enhances the early detection of plant diseases [143]. An electronic nose combined with artificial intelligence successfully detects *G. boninense* in oil palm plantations, offering a reliable and consistent method for early disease detection based on odor parameters [144].

### 6.7. Geospatial Technology

The impact of different oil palm planting densities on BSR disease in Malaysia was examined. Higher-density plots showed increased disease incidence. A spatial analysis revealed moderate-to-weak disease distribution patterns. BSR initially appears randomly and spreads through root-to-root contact. Further research is needed to understand outbreak factors [145]. Drones with cameras were used to monitor Red Ring Disease (RRD) and Bud Rot (BR) in oil palm plantations in Ecuador. Vegetation indices and the Visible Atmospherically Resistant Index (VARI) were computed from captured images, facilitating early detection and response strategies in oil palm cultivation [146]. L-band SAR satellite images can be used to effectively detect BSR disease in oil palm plantations by analyzing backscatter coefficients and structural changes. This method shows promise for efficient BSR detection in oil palm plantations. Higher planting densities in oil palm plantations are associated with an increased incidence of BSR disease caused by *Ganoderma*. A spatial analysis revealed a weak pattern of disease occurrence. Planting density influences the outbreak of *Ganoderma* disease in oil palm plantations [147].

Limited research exists on *Ganoderma* disease’s spatial spreading patterns and distribution within plantations. Understanding disease spread is crucial for effective control measures. A spatiotemporal analysis offers valuable insights into disease epidemiology, identifying transmission factors and informing control strategies. Gathering such information enhances the understanding of disease dynamics and aids in targeted interventions [148]. QuickBird satellite imagery was used to detect and map basal stem rot disease in oil palm plantations. Image segmentation and vegetation indices accurately identified infected areas with 84% accuracy. The study concluded that QuickBird imagery is a reliable method for estimating the extent of basal stem rot disease in oil palm plantations [149]. Advancements in computer technology have simplified plant disease analysis using GIS. GIS helps in understanding disease occurrence and implementing preventive measures. Spatial statistical approaches refine disease hotspot identification and aid in analyzing disease distribution patterns [150].

### 6.8. Machine Learning Techniques and Algorithms

UAV and Pleiades imagery, combined with support vector machine (SVM) classification, enabled the development of an early detection model for BSR in oil palms. UAV imagery achieved the highest accuracy, correctly identifying 64.07% to 64.49% of early *Ganoderma* infections, with an overall accuracy of 68.28%. The detailed information provided by the imagery justified the modest accuracy levels [100]. A prototype device was developed for the initial diagnosis of oil palm diseases. It optimized the setup for recording tree trunk sounds and used a CNN to classify wood-knocking sounds. The CNN-based approach achieved approximately 70–80% accuracy in recognizing *Ganoderma* disease. This method shows promise as an initial diagnostic tool for oil palm disease identification [151]. Machine learning models using hyperspectral imaging were developed to detect *G. boninense* fungus in oil palm seedlings. Significant bands in the near-infrared spectrum were identified and used as input parameters for machine learning models. The coarse Gaussian support vector machine achieved a high F-score of 95.21% and a fast performance time, enabling the early detection of *G. boninense* [152].

A study demonstrated the effectiveness of remote sensing and machine learning in detecting and classifying BSR in oil palm plantations. It found that the oblique random forest (ORF) model with the partial least squares method was the most accurate, while the parallel random forest (PRF) model was the fastest. These insights help select optimal models for differentiating healthy from BSR-infected oil palms, which is crucial for maintaining plantation productivity [153]. WorldView-3 imagery and ML algorithms were used to classify BSR severity in oil palm plantations. Outliers were removed for improved accuracy, and new criteria for identifying disease symptoms were proposed [154]. A decision tree classification approach was proposed to detect BSR disease in oil palm trees using ALOS PALSAR 2 satellite data. The method showed moderate accuracy in distinguishing healthy and *G. boninense*-infected trees, requiring further improvements [155].

UAVs and image processing detect *Ganoderma* disease in oil palm with 62.41% accuracy using adaptive filters and supervised classifiers. This method shows promise for mapping the disease using UAV-based imagery and filters [156]. An ANN analysis of spectroscopic and imagery data accurately detects early-stage *G. boninense* infections in oil palm trees. The green wavelength range and frond number are key indicators. This approach shows promise for cost-effective and accurate BSR disease detection [157]. Machine learning and remote sensing were used to predict and map BSR disease in oil palm. The random forest model achieved 91% accuracy, demonstrating its potential for early detection and disease management in plantations [158].

### 6.9. Mathematical Modeling

Researchers have proposed and compared two spatiotemporal hierarchical Bayesian models to study *G. boninense* infection in oil palm plantations. These models incorporate genetic and spatial components, providing insights into the disease dynamics and offering short-term predictions. They are practical tools for understanding and managing the infection [159]. A mathematical model was developed to estimate yield loss in oil palm due to BSR disease. The model, based on Bayesian Model Averaging (BMA), was created using data from commercial oil palm plots. It provides a useful tool for assessing the economic impact of BSR disease on oil palm plantations [160]. A yield loss model was developed to estimate economic losses from BSR in the oil palm industry. The model used a regression analysis and showed reasonable forecast performance. It helps evaluate control measures and estimate the economic impacts of the disease [161].

### 6.10. Molecular Methods

Molecular techniques (PCR, immunoassays, and nucleic acid hybridization) offer specific and sensitive detection of *Ganoderma*. The ITS regions of rDNA serve as PCR targets. Polyclonal antibodies detect *Ganoderma* proteins for serological assays (ELISA, DIBA). These methods enable early diagnosis in greenhouse and field settings, facilitating disease management [162]. Molecular methods validated *Ganoderma* infections in Indonesian oil palm fields. The *CHALCONE* gene was confirmed as an early infection indicator. The study highlights the importance of specific genes in *Ganoderma* detection [20]. The *ERG11* gene from *G. boninense* was characterized, and its expression during interaction with oil palm was investigated. The findings provide valuable molecular information for controlling basal stem rot in oil palm [163]. Real-time PCR assays were developed to detect *G. boninense* in oil palm. They showed higher sensitivity and speed than conventional methods, successfully detecting the pathogen in inoculated plantlets. This offers promising prospects for effective disease control [164].

*Ganoderma boninense* was analyzed using SSR markers, and genetic diversity was observed within individuals and variations among populations. A phylogeny analysis identified distinct clusters based on origin. These findings inform control strategies for basal stem rot disease [165]. An automated microfluidic device was developed for the rapid DNA extraction and identification of *G. boninense*. The device demonstrated precise dimensions and the successful differentiation of *G. boninense* from another pathogenic fungus. This highlights the potential of microfluidics for the efficient and label-free detection of *G. boninense* [166]. A colorimetric assay was developed for detecting *G. boninense* and other fungi DNA in oil palm. The assay uses DNA–nanoparticle conjugates that aggregate in the presence of complementary DNA. It is rapid, sensitive, selective, and simple to use, making it suitable for field diagnostics without specialized equipment or training [167].

The loop-mediated isothermal amplification (LAMP) method offers fast and sensitive detection of *G. boninense*. Outperforming PCR, it accurately identifies pathogenic strains, promising efficient *Ganoderma* detection [168]. LAMP is a tool for *G. boninense* detection without a thermal cycler. Gel electrophoresis confirms specific gene bands. It shows promise for sensitive detection in oil palm estates [169]. Immunochromatographic techniques detect *Ganoderma* sp. infection in oil palm. Antibodies derived from various isolates show variations in reactivity and sensitivity. Mycelium-derived antibodies perform better, indicating their potential for the early detection of *Ganoderma* sp. infection [170]. The planta infection system was developed to detect *Ganoderma*-induced BSR symptoms in oil palms early. It found that infected plants showed declines in leaf chlorophyll, increased disease severity, and elevated fungal DNA levels. The system also monitored the deterioration of internal stem tissues and changes in phenolic content as part of the plant’s defense mechanism, confirming the system’s effectiveness for early BSR detection [171].

SSRs (microsatellite markers) were developed to study the genetic diversity of *G. boninense*. A set of 17 SSRs was identified and can be used for an effective genetic diversity analysis of this pathogenic fungus. These markers will aid future research on the biology and management of *G. boninense* [172]. Molecular markers for BSR disease in oil palm were identified. Progeny type KA4G1 showed tolerance to BSR, and markers associated with the disease were found. These findings can contribute to developing BSR-resistant oil palm varieties [111]. A protein expression analysis of *Ganoderma*-infected oil palm revealed changes in 51 proteins involved in photosynthesis, carbohydrate metabolism, and immunity/defense mechanisms. These findings provide insights into the molecular responses to BSR disease [173].

A PCR-based diagnostic method was developed to detect *Ganoderma*. Root samples were collected from infected plants, and *Ganoderma* isolates were purified using 2% potato dextrose agar (PDA). Early detection techniques are crucial for effectively managing BSR, which is spreading rapidly in oil palm-growing regions [174]. The root proteins of oil palm seedlings infected with *G. boninense* showed significant changes in abundance compared to healthy seedlings. Specific proteins, including caffeoyl-CoA O-methyltransferase, enolase, fructokinase, and ATP synthase, exhibited notable alterations. These protein changes have implications for managing basal stem rot in oil palm plantations [175]. *Ganoderma* isolates from various locations in India were tested for their pathogenicity on coconut seedlings. DNA-based techniques, including PCR and an RAPD analysis, were used for the early detection of the pathogen. Specific DNA fragments were successfully amplified, indicating the presence of *Ganoderma* [176]. *Ganoderma* species causing stem rot in oil palm trees were examined, with *G. zonatum* being the most prevalent. Genetic heterogeneity was observed within the *Ganoderma* species, emphasizing the importance of *G. zonatum* in disease spread. Multiplex PCR proved effective for species identification [177]. ELISA-PAb, an immunological test, was developed for the early detection of BSR disease in oil palm. It recognized *Ganoderma* species but had cross-reactivity with saprophytic fungi. Although it showed better detection capabilities than cultural-based methods, it lacked specificity for accurate BSR detection. Further research is needed to improve its specificity for precise *Ganoderma* detection [178].

Molecular identification methods were lacking for *G. boninense* isolates in Sabah, Malaysia, causing uncertainty about their aggressiveness. A DNA sequence analysis of *Ganoderma* isolates from the Langkon Oil Palm Estate revealed their similarity to aggressive strains from West Malaysia. Further investigations are needed to understand Sabah’s *G. boninense* isolates [179]. The selection of suitable reference genes for gene expression normalization in *G. boninense* using qPCR involved an analysis of seven potential reference genes, ultimately identifying *β-tubulin*, *eEF2*, and *α-tubulin* as the most stable options, thereby recommending them for accurate normalization in qPCR experiments [180].

A diagnostic method developed for early *Ganoderma* detection in crops using polyclonal antisera showed efficacy with a 1:1000 titer value and clear color distinctions in the DIBA test. ECP antisera detected positive reactions in infected plant parts. ELISA and DIBA tests have the potential for large-scale screening and early detection, aiding in effective management strategies against *Ganoderma* disease in palm crops [181]. Two types of antisera were generated for *Ganoderma* detection. The DIBA test effectively differentiated healthy from diseased samples. The antiserum developed from purified protein had a low cross-reaction, while the crude extract antiserum showed a higher cross-reactivity. MS antiserum detected infected roots and field palms, while SE antiserum detected various infected samples. ELISA and DIBA tests were useful for early *Ganoderma* detection in palm crops [182].

RAPD and PCR-RFLP techniques were used to characterize *Ganoderma* isolates from different hosts. Consistent patterns were observed within the same species, and a cluster analysis showed host-specific clustering. These techniques effectively distinguish *Ganoderma* species in Peninsula Malaysia [183]. Serological techniques (ELISA, DIBA) using specific antibodies successfully detected *Ganoderma* in infected coconut plantations, offering the potential for early BSR disease diagnosis [184]. In surveyed oil palm gardens, 46 suspected BSR cases caused by *Ganoderma* were found. PCR confirmed 21 positive isolates with diverse morphological characteristics and distinct groups. Unique features and a spore-forming ability were observed in some isolates [185].

MAbs were developed against *G. boninense*, and *G. boninense* mycelium extract was used as the immunogen. Cross-reactivity with other fungi was observed initially, highlighting the need for improved specificity. Future research will focus on reducing cross-reactivity and optimizing *G. boninense*-specific MAbs through a co-immunization program [186]. A PCR-RFLP analysis with specific primers distinguishes pathogenic *Ganoderma* species causing BSR in oil palms. It provides a reliable method for identification, confirming the specificity of oil palm *Ganoderma* strains. This protocol can be used as a standard for *Ganoderma* detection in oil palm [187]. The GanET PCR primer amplifies *G. boninense* DNA. A DNA extraction and PCR method detects *G. boninense* in oil palm tissue. *Ganoderma* DNA was detected in frond bases, correlating with basal stem rot development in young palms. Infection of lower frond bases increases rot likelihood [188].

The development of an immunoassay-based detection kit for oil palm was reviewed. The review covers the importance of detection tools, specifications for new detection products, and the use of antibodies for detecting antigenic material from dried oil palm leaf samples [189]. The researchers investigated *Ganoderma* species causing stem rot in oil palm trees, with *G. zonatum* being the most common species. They observed genetic diversity within the *Ganoderma* species, highlighting the importance of *G. zonatum* in the spread of the disease. For species identification, the researchers utilized multiplex PCR, a molecular method that effectively distinguishes *Ganoderma* species [190]. The rapid molecular diagnostic method developed for detecting pathogenic *Ganoderma* species in oil palm using the ITS region was used at Papua New Guinea Oil Palm Research Association with EU-STABEX funding. The organism was found in a saprobic state on deceased coconut palms [191]. Investigating *Ganoderma* species in oil palms in Papua New Guinea involved using molecular methods to study pathogen populations and the sexual cycle. Random amplified polymorphic DNA and conserved genomic markers were utilized. These methods are valuable for analyzing and identifying *Ganoderma* populations in oil palm and other affected crops [192]. Molecular methods for the detection of *Ganoderma* and their corresponding primer sequences are listed in Table 1.

#### Pathogenicity-Related Molecular Methods

Pathogenicity is a critical aspect of understanding the interactions between *G. boninense* and oil palm plants [11]. BSR disease, caused by *G. boninense*, poses a significant threat to oil palm plantations in Southeast Asia [51]. Several studies have been conducted to investigate the pathogenicity mechanisms, molecular characteristics, and genetic makeup of *G. boninense*. These studies have explored various aspects, including genomics [14], metabolomics [21], soil properties [16], gene expression [89,195], transformation protocols [196,197], and the role of specific genes and enzymes [175,198,199]. By gaining insights into the pathogenicity of *G. boninense*, researchers aim to develop effective strategies for disease control and to identify potential resistance genes in oil palm [200]. The following part contains several research studies conducted on the pathogenicity of *G. boninense*. 

A comparative genomics analysis was conducted on ten *Ganoderma* species/strains, identifying *G. boninense* as a hemibiotrophic fungus with specific pathogenic traits. Focusing on key genetic elements, such as CAZymes and secondary metabolite core genes, the study revealed unique genes and potential effectors in *G. boninense* associated with plant pathogenesis and wood degradation. These findings advance our understanding of *Ganoderma* species and their ability to produce bioactive secondary metabolites, shedding light on their pathogenic mechanisms [15]. Researchers used LC-MS/MS and GC-MS to analyze the metabolites released by *G. boninense* during its interaction with oil palm plantlets. They identified bioactive compounds such as azelaic acid, isonicotinic acid, and ricinoleic acid. The detection of ricinoleic acid provides insights into the pathogenesis of basal stem rot [201].

The role of the *NRPS* gene in *G. boninense* was explored and the research identified the core motifs of the *NRPS* gene and predicted the synthesis of siderophores, indicating their involvement in the pathogenicity of *G. boninense*. The upregulation of the *GbNRPS* gene in susceptible oil palm clones infected with the fungus suggests its significance in disease development [202]. The genome of *G. boninense* T10 was sequenced and analyzed, and a comparative genome analysis identified candidate effector proteins (CEPs) associated with carbohydrate metabolism and host defense suppression, providing insights into the pathogenicity of *G. boninense* [203]. The study also revealed the genome architecture and the absence of preferential association of *CEP* genes with transposable elements.

The involvement of GbHog1 MAPK in the response of *G. boninense* to salinity stress was investigated. They identified a full-length cDNA encoding the Hog1-type MAPK and observed its upregulation in high salt concentrations. These findings contribute to our understanding of the pathogenicity mechanisms of *G. boninense* and its response to environmental stressors [204]. Investigating various *Ganoderma* species’ pathogenicity on young, healthy landscape trees revealed that most species did not harm the trees. Only *G. sessile* showed some infectivity in healthy sapwood. Further research is required to explore alternative infection routes and understand *Ganoderma* species’ impact on older trees with heartwood [55]. A highly efficient and reproducible Agrobacterium-mediated transformation protocol for *G. boninense* was developed. The protocol successfully transformed various types of explants, achieving a maximum efficiency of 62% using protoplasts and the Agrobacterium strain LBA4404. The transformed fungi expressed marker genes, allowing for the study of *G. boninense* pathogenicity and its interaction with oil palm roots [196].

A study analyzing the transcriptomes of three isolates of *G. boninense* was conducted with varying pathogenicity levels. The research focused on genes involved in lignin degradation, including laccase genes, to understand the infection process. The study identified 33 *laccase* genes and revealed differential expression patterns among the isolates, with the most pathogenic isolate exhibiting the upregulation and downregulation of unique transcripts. The findings shed light on the gene expression profiles and potential mechanisms underlying *G. boninense* infection in oil palm [205]. An artificial infection assay was developed for BSR caused by *G. boninense* in oil palm. Infected plants displayed characteristic symptoms and a reduced lignin content. The study sheds light on BSR and provides a valuable tool for further research and management of the disease in the oil palm industry [206]. A study was performed on *G. boninense*, revealing its epidemiology, pathogenicity, and genetic diversity. The research highlighted the importance of close contact for infection, identified basidiospores as a critical factor, and detected genetic diversity within plantations. The study also identified enzymes and potential toxins involved in pathogenesis. These findings contribute to disease control and may assist in identifying resistance genes through a transcriptomic analysis in oil palm [207]. The infection potential of *G. boninense* isolates in causing BSR disease in oil palm was investigated. Isolates with strong antagonistic reactions and high ligninolytic enzyme production were found to cause high disease severity when infecting oil palm seedlings [135].

Researchers focused on the genetic characteristics of *G. boninense* and they targeted the cyclophilin-encoding gene and successfully amplified a DNA fragment, showing similarities to other plant pathogenic fungi. The study provides insights into the genetic makeup of *Ganoderma* fungi and their involvement in BSR disease in oil palm [208]. Researchers investigated gene expression in oil palm seedlings infected with *G. boninense* and the symbiotic fungus *Trichoderma harzianum*. Gene expression patterns differed between the interactions, with upregulation and downregulation observed in response to *G. boninense*, while gradual upregulation occurred during symbiotic interactions with *T. harzianum*. The findings indicate the plant’s ability to distinguish between pathogenic and symbiotic interactions and suggest gene expression as a factor in resistance and growth promotion [64].

The *PGIP* gene (*EgPGIP*) in oil palm inhibits fungal infection but is downregulated during *G. boninense* infection. This study provides insights into the molecular interactions between oil palm and the fungal pathogen, contributing to disease resistance research [63]. *Ganoderma boninense* infects oil palm roots, exhibiting penetration and rapid colonization of the lower stem. The fungus behaves as a hemibiotroph initially, occupying host cells, and later transitions to inflict a necrotrophic attack on cell walls. Melanized mycelium formation surrounds the roots, leading to multiple infections in mature palms. This study provides insights into the infection process and spread of *G. boninense* in oil palm [209].

### 6.11. Multispectral and Hyperspectral Data Analyses

A spectral analysis is crucial for managing *Ganoderma* disease in oil palm plantations. While it does not prevent pathogen infiltration, it enables early detection and intervention to limit spread. Identifying the specific spectral signatures of *Ganoderma* infection allows for timely control measures, minimizing economic losses. Early detection is key, facilitating prompt action and guiding sustainable practices. While it does not stop pathogen entry, a spectral analysis enhances plantation resilience and productivity against *Ganoderma*. NIR spectroscopy detects *G. boninense* infection in oil palm trees by analyzing spectral reflectance changes around 1450 nm. Accurate differentiation is achieved using chemometric and machine learning techniques, showcasing the potential for non-destructive detection [210]. The early detection of *G. boninense* infections in oil palm seedlings was studied using VIS-NIR hyperspectral images. Significant bands were identified, and a linear SVM achieved an accuracy of 94.8% using a single-band reflectance at 934 nm. This has potential benefits for the oil palm industry by enabling the early detection and prevention of disease spread [211]. The automation of BSR detection in oil palm seedlings using deep learning and hyperspectral images was investigated. The VGG16 model trained with original images achieved high accuracy (91.93%), eliminating the need for manual image attribute extraction and improving efficiency in BSR detection [212]. Automatic BSR detection in oil palm trees using hyperspectral data and deep learning techniques offers high accuracy and fast classification, addressing labor shortage challenges and preventing disease spread [212].

Airborne remote sensing combined with ground detection improves *Ganoderma* disease control in oil palms by enhancing hyperspectral signatures and reducing interference and noise. This approach offers a higher accuracy than traditional methods [96]. Satellite data detected *Ganoderma* attacks on oil palm trees in South Sumatra, Indonesia. Chlorophyll content and NDVI showed a negative correlation with *Ganoderma* severity. A decrease in NDVI indicated the occurrence and severity of the attacks. These findings can aid in monitoring and managing *Ganoderma* infestations in oil palm plantations [213]. UAV-based remote sensing detects *Ganoderma*-infected oil palms, aiding in the early detection of BSR disease. An Artificial Neural Network (ANN) model achieves accurate classification using UAV imagery. This cost-effective approach enables the efficient monitoring of BSR in oil palm plantations [214].

NIRS combined with machine learning offers a promising approach for early *G. boninense* detection. NIRS provides accurate and non-destructive measurements, while machine learning enables efficient analysis and predictive capabilities. This approach can potentially create an effective and environmentally friendly system for the early detection of *G. boninense* [215]. Aerial remote sensing with multispectral, hyperspectral, and radar sensors is effective for detecting *Ganoderma* disease and bagworm infestation in oil palm plantations. It provides fast and efficient detection for large-scale plantations, with high accuracy (>80%) using UAV-based multispectral remote sensing [216]. Hyperspectral remote sensing was used to analyze spectral reflectance differences between healthy and *G. boninense*-infected oil palm seedlings’ young leaves. Spectral signatures were extracted and averaged, revealing noticeable differences, particularly in the near-infrared region. This suggests the potential of using F1 spectral reflectance for the early detection of *G. boninense* infection [217].

Multispectral remote sensing and OBIA are used to detect *Ganoderma* infection in oil palms. High accuracy was achieved (91.8%) with specific methods and SVM classifier. There is potential for detecting moderate and severe infection, but advancements are needed for early detection [218]. Detecting early *G. boninense* infections in oil palm seedlings using hyperspectral images revealed significant bands for distinguishing infected and uninfected seedlings. SVM models achieved 100% accuracy using selected bands. Combining frond 1 and frond 2 data improved accuracy, indicating that aerial images can be used to detect infections without separating fronds during preprocessing [97]. Spectroscopy with dimensionality reduction accurately classifies BSR severity in *Ganoderma*-infected oil palm. A dielectric spectral analysis (QDA and PCA) achieves up to 96.36% accuracy. SVM without reduction has a lower accuracy (79.55%). Dimensionality reduction is crucial for precise BSR classification [219].

Hyperspectral remote sensing holds promise for early *Ganoderma* BSR disease detection, but cost is a challenge. UAV-based multispectral indices show moderate accuracy. Further research aims to develop a comprehensive disease severity index for large-scale management [220]. Airborne hyperspectral remote sensing detects *Ganoderma* disease severity in oil palm with moderate accuracy. Research is ongoing to enhance detection methods [221]. Impedance measurements showed high accuracy in classifying *Ganoderma* BSR disease severity levels in oil palm. The findings highlight the potential of electrical properties, specifically impedance, for early disease detection using spectroscopy techniques [123]. UAV multispectral images and vegetation indices were used to efficiently detect *G. boninense* infection in oil palm trees, allowing for the mapping of the basal stem rot (BSR) distribution [222].

Spectral indices were developed to detect *Ganoderma* disease in oil palm seedlings. Six effective indices, including “Ratio 3”, were identified, demonstrating potential for the early detection of the infection [223]. Hyperspectral remote sensing and UAV systems were tested for early BSR disease detection in oil palm. The results showed potential, but improvements are needed for accuracy. GIS integration and disease census-based models shows promise in the monitoring of disease spread [224]. Airborne hyperspectral remote sensing was used to detect BSR disease severity in oil palm. SRI and EVI indices showed moderate accuracy, while other indices performed poorly. CR techniques revealed significant distinctions among severity classes. Further research is needed for specific BSR detection methods in oil palm using hyperspectral remote sensing [225]. WSNs have potential in palm oil plantations for *Ganoderma* detection. Feature selection using a rough set technique and a genetic algorithm improves efficiency in selecting relevant soil features for data transmission [226]. Fourier transform infrared spectroscopy (FTIR) shows promise as a detection tool for *G. boninense* in oil palm trees. FTIR results exhibited noticeable differences between infected and healthy trees, resembling the FTIR characteristics of pure *G. boninense*. This suggests the potential of FTIR for detecting fungal infection in oil palm trees [227].

Fourier transform infrared (FT-IR) spectroscopy was utilized to develop a statistical model for detecting *Ganoderma*-infected oil palm trees. The model achieved high accuracy in differentiating healthy from infected trees, demonstrating potential for early disease detection and reduced chemical treatments [228]. Hyperspectral field data and vegetation indices classify *Ganoderma* disease stages in oil palm plants. The Modified Red-Edge Simple Ratio (MSR705) effectively distinguishes healthy, semi-healthy, and severely damaged plants, aiding disease monitoring and management [229]. Airborne hyperspectral imaging was employed to detect *Ganoderma* disease in oil palm crops. Support vector machines (SVMs) and tree crown detection methods were utilized for the accurate classification and identification of infected oil palms. The results demonstrated improved classification accuracy using SVM and the enhanced detection of *Ganoderma*-infected trees [230,231]. The detection and monitoring of *Ganoderma* stem rot and bagworm infestation in oil palm crops are improved through spectral data analysis. This provides valuable insights for the effective management and assessment of damage in oil palm trees [231,232]. A spectral analysis was used to detect *Ganoderma* stem rot and bagworm infestations in oil palm trees. Bagworms were detected at 570 nm, 680 nm, 734 nm, 787 nm, 996 nm, and 1047 nm. *Ganoderma* disease was identified at 515–586 nm, 615–622 nm, 633–644 nm, 690–708 nm, 727–758 nm, and 773–784 nm [231].

FTIR spectrometry accurately detects *Ganoderma*-infected oil palm trees at different stages of infection. This method accurately distinguishes healthy and infected leaves, highlighting its potential for the early detection of *Ganoderma* disease in oil palm plantations [233]. Portable hyperspectral remote sensing detected *Ganoderma* disease in oil palm trees by analyzing the reflectance spectra of infected leaves. Significant bands and wavelengths were identified for spectral discrimination. Classification using a maximum likelihood classifier showed the potential of hyperspectral remote sensing for identifying and classifying *Ganoderma* disease in oil palm trees [234]. Ultrasonic measurements revealed a reduced density and lower ultrasonic wave velocities in oil palm trunks infected with *G. boninense* disease. These findings aid in detecting affected areas [235]. Hyperspectral reflectance data were used to detect *Ganoderma* disease in oil palm trees non-destructively. A statistical model effectively differentiated between disease stages with high accuracy. Canopy reflectance yielded valuable information for distinguishing healthy and sick trees. Potential applications and areas for improvement were also discussed [236]. 

High-resolution field spectroradiometers were used to detect *Ganoderma* disease in oil palms. New reflectance indices were created and evaluated, showing the best clustering results for different infection levels. A specific spectral index using wavelengths at 610.5 nm and 738 nm demonstrated potential for the early detection of *Ganoderma* in oil palm [237]. Airborne hyperspectral imagery is effective for the timely detection of *Ganoderma* disease in oil palm plantations. It provides an accurate assessment using vegetation indices and red-edge techniques. Accuracy ranges from 73% to 84%, with red-edge-based methods performing well [238]. Hyperspectral data used to detect oil palm disease, improve accuracy with wavelet-based techniques, and develop new vegetation indices and red-edge techniques enable the large-scale mapping of infection [239]. *Ganoderma* disease in oil palm trees was detected and mapped using airborne hyperspectral imagery. A vegetation analysis of preprocessed data revealed high accuracy with a Red-Edge NDVI 705 and VOG1 composite. Red-edge-based indices are recommended for disease detection in oil palm [240].

Hyperspectral reflectance data were investigated as a non-destructive method for detecting *Ganoderma* disease in oil palm. A statistical model was calibrated using field measurements, achieving high accuracy in classifying disease severity. The method shows potential for efficient and cost-effective diagnosis in oil palm plantations [241]. Polyclonal antibodies (PAbs) and PCR tests were used to detect *Ganoderma* disease in coconuts. PAbs showed an early detection capability, while PCR confirmed the presence of *Ganoderma*. This method aids in early disease detection in coconuts [242]. Portable hyperspectral remote sensing was used to detect *Ganoderma* infections in oil palm trees. Reflectance and derivative spectra were analyzed to identify spectral differences between healthy and infected palms. Derivative data showed more significant differences, indicating the potential for detecting *Ganoderma* infections. Early detection, however, remains challenging [243].

### 6.12. Image Processing

Image processing and Artificial Neural Network methods were employed to detect *Ganoderma* infection in oil palm plantations using drone-captured images. The model achieved 73.8% accuracy and provided a distribution map of the infection. This non-destructive approach aids in the early detection of basal stem rot disease, contributing to the management of oil palm plantations [244]. Thermal imaging is used with machine learning algorithms to distinguish healthy and BSR-infected trees. Imbalanced data are handled, and evaluation metrics are used to assess algorithm performance. The study aims to provide a user-friendly approach for accurate BSR disease prediction using machine learning [245].

Thermal imaging shows promise for the early detection of BSR in oil palm seedlings. By analyzing thermal properties and using a statistical analysis and classification models, an accuracy of 80.0% is achieved with support vector machines (SVMs). This underscores the potential of thermal imaging as a tool for BSR detection in oil palm seedlings [246]. ALOS PALSAR-2 images detected *G. boninense* in oil palm trees. The HV backscatter variable achieved high success rates. MLP and RF models showed accuracies of 95.65% and 92.70% respectively, using HV polarization. This is valuable for predicting BSR disease in oil palm trees [247]. High-resolution imagery and Faster-RCNN enable the automatic detection and classification of oil palm trees. Key characteristics aid accurate detection, and Resnet-50 shows superior performance. This is promising for enhancing oil palm tree management [248].

### 6.13. Spectral Index Analysis

Visible spectral indices from aerial photographs accurately differentiate the severity levels of *G. boninense* infection in oil palm plantations. ExG, ExR, and CIVE indices effectively assess infection levels in individual trees. This method provides an alternative approach for evaluating *G. boninense* infection using visible spectral indices from aerial photographs [249]. A spectroscopic analysis enables the accurate and early detection of *Ganoderma* disease in oil palm, with a high classification accuracy of 97%. It proves effective for the rapid screening and detection of basal stem rot (BSR) in oil palm [250].

### 6.14. Systematic Screening

A methodology for the early screening of *Ganoderma* disease susceptibility/tolerance in a pot–tray system was introduced. Controllable parameters were optimized, and the screening method successfully differentiated between highly susceptible and less susceptible progenies. This approach is valuable for assessing the response to *Ganoderma* infection in planting material [163].

### 6.15. Tomography Techniques

Electrical impedance tomography enables the early identification of BSR disease in oil palm trees by comparing tomograms and establishing a range of electric impedance values. This method effectively detects the disease, even in asymptomatic trees, facilitating proactive disease management approaches [251]. Tomography shows promise as an effective tool for accurately and efficiently managing *Ganoderma* infection in oil palm plantations [252]. A system was developed to detect and localize *Ganoderma* infection in oil palm stems using tomography images. Expert rules based on lesion recognition were used for automatic detection. A fuzzy inference system classified lesion patterns and assigned probabilities of infection. The study also explored combining data from multiple lesions to assess overall tree health [253].

*Ganoderma* stem rot in oil palm plantations is challenging to detect. A portable gamma-ray computed tomography system offers an efficient on-site solution for distinguishing healthy and infected stems, aiding in the assessment of basal stem rot damage [254,255]. Tomography images are used to automatically identify *Ganoderma* infection in oil palm stems. A fuzzy inference system incorporating expert knowledge is employed for pattern classification. Several combinations of fuzzy inference methods yield promising results in approximating expert estimations [256].

### 6.16. Terrestrial Laser Scanning

Terrestrial laser scanning identifies *Ganoderma*-induced basal stem rot (BSR) in oil palm trees. Effective indicators include crown strata number 17 and crown area. Monitoring every 4 months detects BSR-infected trees, preventing crop losses and enabling early treatment [257]. Ground-based LiDAR (terrestrial laser scanning) analyzed oil palm tree canopy architecture and classified trees based on basal stem rot (BSR) severity. The frond number was effective for early detection. A linear model combining the frond number, frond angle, and canopy strata achieved accurate classification. TLS and point cloud data offer promising BSR detection in oil palm trees [258]. TLS technology was used to detect BSR disease in oil palm trees caused by *G. boninense*. A crown strata analysis accurately classified healthy and unhealthy trees with a 92.5% accuracy rate. TLS is effective for BSR detection in oil palm plantations [259]. ML with TLS data accurately classified different levels of BSR disease in oil palm trees. The KNB model achieved 85% accuracy and a Kappa coefficient of 0.80, successfully predicting early BSR infection. ML and TLS prove effective for disease classification in oil palm trees [260]. Terrestrial laser scanning (TLS) accurately analyzes oil palm trees and identifies parameters related to basal stem rot (BSR) severity. The frond number proves effective in distinguishing severity levels. TLS enables the early detection of BSR and supports disease management in oil palm plantations [261].

## 7. Research on Disease Management Strategies to Control Basal Stem Rot

BSR caused by *Ganoderma* is a major disease affecting oil palm production. Effective disease management strategies are crucial for controlling the spread and minimizing the impact of BSR. Cultural practices, biological control, chemical control, host resistance, and soil management are some key disease management strategies employed to control the BSR of *Ganoderma* [2,61]. However, to manage BSR, an integrated *Ganoderma* management (IGM) approach has been recommended, involving sanitation, biological agents, fertilizers, and chemical control [262]. In the following section, we discuss various disease management strategies for controlling basal stem rot (BSR) caused by *Ganoderma* in oil palm plantations.

### 7.1. Agricultural Practices

Co-infection with different strains of *G. boninense* in oil palm seedlings results in similar disease symptoms but decreased severity compared to a single aggressive strain. Co-infection shows higher fungal colonization in oil palm roots. These findings suggest a potential impact of strain co-infection on disease development [263]. A field trial in an Indonesian oil palm plantation studied different practices to reduce *Ganoderma* infection in replanted oil palms. The results showed that fallowing and poisoning, combined with the windrow 2:1 system, effectively reduced infection rates to 3%. Fallowing without poisoning in the windrow 2:1 system also resulted in a low infection rate of 6% [264].

### 7.2. Cultural Practices

Land management methods, including fallowing and proper plant placement, were found to be effective in controlling basal stem rot disease caused by *Ganoderma* in oil palm. Fallowing reduced disease incidence, while avoiding infected remnants and exposing soil to sunlight aided in disease reduction [265]. Crop rotation with sugarcane was studied as a cultural practice to combat BSR disease in oil palm production. The research explored the biological and molecular characteristics of the converted land and demonstrated the potential of crop rotation to manage BSR while maintaining productivity [266]. Mixed planting with edible herbaceous perennial plants (canna, arrowroot, cocoyam, and water yam) suppressed the growth of oil palm seedlings, but it had a minor effect on *Ganoderma* infection. Arrowroot showed the strongest competition, followed by canna and cocoyam, while water yam had minimal interference [267].

A cost-effective method used to limit *Ganoderma* infection in oil palms on peat involved creating trenches around infected palms, isolating a designated area. This approach effectively minimized the spread to healthy neighboring palms. Additionally, using the excavated soil to mound the base of the infected palm extended its productive lifespan [268].

Field experiments assessed the effectiveness of cultural practices in controlling BSR in oil palm. Methods such as sanitation, hole-in-hole planting, digging and mounding, and isolation trenches showed promise in preventing early-stage infection and prolonging palm lifespan [269].

Sanitation plays a crucial role in managing BSR caused by *Ganoderma* sp. in oil palm plantations. Old palm remnants can serve as sources of disease inoculum during replanting. Factors like palm size, growth vigor, and inoculum size impact disease development. Controlling BSR early and preventing root contact with infected palm trunks are vital [270]. Research showed that poisoning old palms before felling them significantly reduced *Ganoderma* disease in oil palm replanting. By comparing different replanting methods, it was found that there was no significant difference in disease levels between underplanting and complete felling methods. The study suggests that poisoning old palms prior to felling is an effective cultural practice for disease management [271].

### 7.3. Biological Control

#### 7.3.1. *Trichoderma*

*Trichoderma* spp. are effective biocontrol agents against plant diseases, including BSR caused by *Ganoderma* spp. in oil palm. They employ mechanisms such as mycoparasitism, antibiotic production, enzyme secretion, nutrient competition, and induced plant resistance. Understanding these mechanisms can enhance *Trichoderma*’s effectiveness as a biocontrol agent [19]. *Trichoderma* spp. strains TSU and TGLP showed strong inhibition of *G. boninense* and produced chitinase, glucanase, and indole acetic acid. Molecular identification confirmed their potential as biocontrol agents for basal stem rot in oil palm [272,273].

*Trichoderma asperellum* shows high potential as a biocontrol agent against *Ganoderma* species, effectively inhibiting their mycelial growth [274]. Endophytic *Trichoderma* isolates, particularly *T. asperellum* M103 and *T. harzianum* M108, show strong potential as biocontrol agents against *G. boninense*, offering promising options for managing BSR disease in oil palm cultivation [275]. Potential soil microbial antagonists, *Trichoderma harzianum* (AC2) and *Burkholderia gladioli* (N1), have been identified for BSR control in oil palm [276]. 

*Trichoderma harzianum* was tested as a biological agent to control *G. boninense* in palm oil plants. Different media compositions were evaluated, showing significant effects on spear leaf count, mycelium growth, and root length. The combination of 25% oil palm empty fruit bunches and 75% cow dung compost yielded the best results in reducing spear leaf count, while 50% OPEFB and 50% cow dung compost stimulated root growth [277]. Endophytic *Trichoderma* species from North Sumatra, Indonesia, were evaluated for their effectiveness in preventing BSR disease in oil palm nurseries. The application of these species, particularly *T. reesei* ET501, *T. asperellum* ET537, and *T. asperellum* ET523, effectively suppressed *G. boninense* infection. The treatment also led to an increase in total phenolic content (TPC), enhancing the resistance of oil palm seedlings to BSR disease [278]. *Trichoderma virens* was studied for its efficacy in controlling *G. boninense* in oil palm pre-nurseries on peat medium. The experiment utilized various doses of *Trichoderma virens* and assessed several parameters. The results showed that *Trichoderma virens* effectively reduced the disease intensity caused by *G. boninense* and promoted stem diameter growth in oil palm seedlings [279]. Pre-inoculating oil palm seedlings with *T. harzianum* and/or *Bacillus cereus* effectively suppresses *Ganoderma* disease, promoting seedling growth and reducing disease severity. The combination of *T. harzianum* and *B. cereus* shows promising potential for the sustainable control of *Ganoderma* in the oil palm industry [19].

*Pseudomonas aeruginosa* and *T. asperellum* demonstrate antagonistic properties against *G. boninense* and promote plant growth. They inhibit *G. boninense* growth and exhibit positive traits like phosphate solubilization and indole acetic acid production. *Trichoderma asperellum* also shows siderophore production, and both BCAs produce hydrolytic enzymes [117]. The metal-tolerant endophytes Diaporthe miriciae LF9 and *Trichoderma asperellum* LF11 inhibit *G. boninense* (GB) in vitro. *Diaporthe miriciae* LF9 improves tolerance and reduces disease severity in oil palm seedlings, while *T. asperellum* LF11 is less effective. These endophytes have potential for controlling GB in metal-laden soils [280]. Endophytic *Trichoderma* species from oil palm roots, including *Trichoderma reesei* ET501, show potential as control agents against *G. boninense*. *Trichoderma reesei* ET501 exhibits high antagonistic and antibiosis activity, making it a promising candidate for oil palm disease management [281]. Standardized methods for evaluating the effectiveness of *Trichoderma* isolates against *Ganoderma* in seedlings are presented. External and internal assessments monitor mycelium, basidiocarp, foliar symptoms, seedling mortality, and decayed tissues. These methods enable the timely detection of *Ganoderma* symptoms and a reliable evaluation of *Trichoderma* efficacy in controlling *Ganoderma* [282]. The use of biological agents derived from Indonesian tubers was explored to control *Ganoderma* stem rot in oil palm. Isolates from Belitung tubers exhibited high antagonistic activity against *Ganoderma*, particularly *Trichoderma* species. These findings suggest the potential of these endophytic fungi as biological agents for controlling stem rot disease in the oil palm industry [283].

*Trichoderma* modulates genes related to hormone production, antioxidant systems, and cell wall metabolism. *Trichoderma* also acts as a biofertilizer, improving nutrient status. The research enhances our understanding of *Trichoderma*’s biocontrol potential against *Ganoderma* in oil palm [284]. *Trichoderma* spp. showed promise as a biological control agent for basal stem rot in oil palm. The mixture of three *Trichoderma* species reduced disease symptoms, enhanced plant defense mechanisms, and slowed disease progression. These findings suggest *Trichoderma* spp. as a sustainable solution for managing basal stem rot in oil palm [285]. Weed species were analyzed as potential markers for *Ganoderma* presence in oil palm plantations. *Cyclosorus aridus*, *Cyperus rotundus*, and *Stenochlaena palustris* were found in infected areas. Temperature and moisture conditions were recorded [286]. Two strains of *Trichoderma harzianum* (FA 1132 and FA 1166) were tested as biocontrol agents against basal stem rot in oil palm seedlings infected with *G. boninense*. FA 1132 showed the highest effectiveness in reducing disease severity, while FA 1166 and the mixture of both strains were ineffective [287]. *Trichoderma harzianum* T32 exhibited mycoparasitic behavior against *G. boninense* upm001, leading to the deformity and shrinkage of *Ganoderma* mycelia. The expression of the chitinase gene was detected during this interaction, emphasizing its role in *Trichoderma*’s biocontrol activity against *G. boninense* [288]. *Trichoderma virens* 159c suppressed *G. boninense*. Ethyl acetate extract displayed strong antifungal activity, deforming *G. boninense* mycelia. Specific fractions containing acetamide, alcohol, lactones, and free fatty acids showed high inhibition, including the unique compound PEA. *T. virens* and its compounds hold promise as biocontrol agents against *G. boninense* [289].

TR1, a biocontrol product with *Bacillus* spp. and *Trichoderma* spp., effectively controlled *G. boninense* in oil palm. It reduced colonization, pathogen re-isolation, and ergosterol content in field experiments. TR1 disrupted and lysed mycelium, containing antimicrobial compounds that inhibited *G. boninense*. It shows promise as an eco-friendly alternative to fungicides for *Ganoderma* control [108]. *Ganoderma* sp. grew faster on PDA media, while *Trichoderma* spp. exhibited different growth rates on PDA and MEA media. *Trichoderma harzianum* showed the highest inhibition of *Ganoderma* sp. isolates [290]. 

*Trichoderma harzianum* successfully suppressed *G. boninense* growth in both in vitro and in vivo experiments. Treated oil palm plants remained disease-free, while control plants showed symptoms. *Trichoderma harzianum* holds promise as a biocontrol agent against *G. boninense* in oil palm [291]. The *Trichoderma* strains *T. asperellum* T9 and *T. virens* T29, in combination with palm press fiber (PPF) mulch, successfully controlled *G. boninense*-induced BSR in oil palm seedlings. The treatments delayed disease onset by eight weeks compared to the control, highlighting their potential as biocontrol measures for managing BSR [292]. The *Trichoderma* isolates T9 and T29 effectively suppressed BSR disease. Treatment with *Trichoderma* and PPF mulch delayed BSR onset by eight weeks compared to the control, demonstrating potential biocontrol for managing BSR in oil palm seedlings [292].

The *Trichoderma harzianum* strain FA 1132 effectively controlled BSR disease in oil palm. Application via *Trichoderma*-incorporated surface mulch resulted in symptom-free plants with lower disease severity and increased root and leaf weights. It offers an eco-friendly alternative to chemical pesticides [92]. PSA culture media enhanced the spore production and antagonistic activity of *T. harzianum* and *T. virens* against *Ganoderma*, suggesting its importance in optimizing biocontrol efficacy [293]. *Trichoderma harzianum* treatment enhanced chitinase activity in oil palm tissues infected with *G. boninense*, indicating its potential role in enhancing defense mechanisms against microbial pathogens [294].

The basal rot diseases caused by fungi affect tree vegetation in India’s arid region. *Ganoderma lucidum*, *Inonotus hispidus*, *Phellinus pachyphloeus*, and *P. badius* are common culprits. Decay primarily targets older wood, leading to increased mortality in *Prosopis* and *Acacia* trees. Biocontrol using *Trichoderma* and *Gliocladium* fungi is effective in reducing *G. lucidum* infections [295]. *Trichoderma* inoculation improved seedling growth and establishment in *Acacia mangium*, outperforming fungicide sprays. A new *Trichoderma* production facility was established, resulting in estimated economic benefits of USD 1.5 million annually [296]. *Trichoderma harzianum* showed potential as a biological control agent against *G. boninense*, the pathogen causing basal stem rot in oil palms. Several isolates of *T. harzianum* inhibited the growth of *G. boninense*. The most promising isolate, FA 30, exhibited strong inhibitory properties through the production of antifungal compounds. *Trichoderma harzianum* could be an effective biocontrol agent for managing *G. boninense* infections in oil palms [297]. *Trichoderma harzianum* FA 1132 controlled *Ganoderma* infection in oil palm seedlings, reducing disease severity and the number of infected plants. It shows potential as a biocontrol agent for *Ganoderma* in oil palm plantations [298].

The application of biocontrol agents (*Pseudomonas fluorescens*, *Trichoderma viride*, and *T. harzianum*) with chitin in *Ganoderma*-infected coconut roots induced defense enzymes and phenolics. Increased activities of peroxidase, polyphenol oxidase, phenylalanine ammonia-lyase, chitinase, and beta-1,3-glucanase were observed. The findings suggest enhanced defense mechanisms against *Ganoderma* disease [299]. *Trichoderma* is a biocontrol fungus that parasitizes plant pathogens. It produces cell wall-degrading enzymes, including endochitinase, which effectively suppresses various plant pathogenic fungi, including *Ganoderma* spp. [300]. IOPRI is researching biological control agents and resistant plants for disease management. Promising agents include *Trichoderma* spp., *Gliocladium* viride, *Pseudomonas fluorescens*, and *Bacillus* spp. Field trials showed *Trichoderma* spp. and *G. viride* to be more effective. IOPRI is also exploring methods to enhance agent activity and degrade plant material to eliminate disease sources [301]. *Trichoderma harzianum* effectively controls BSR by significantly reducing disease severity. Early treatment yields better results, with a DSI of 5%, compared to delayed treatment (DSI of 32.5%). The regular application of *T. harzianum* conidial suspensions shows promise for BSR management, offering an alternative to chemical treatments [302]. Native *Trichoderma* spp. biocontrol agents (*T. viride*, *T. harzianum*, and *T. hamatum*) were evaluated for their compatibility with agrochemicals and effectiveness against *Ganoderma* spp. Talc formulations were developed, and neem cake was identified as a suitable substrate. Field trials showed promising results with *T. hamatum*, *T. harzianum*, and *T. viride* formulations combined with neem cake [303].

#### 7.3.2. Other Microbes and Macrofungi

Biological products, though typically applied superficially, possess mechanisms to combat *Ganoderma*, even if it penetrates tree trunks [304,305]. These products include natural antagonists such as endophytic bacteria, mycorrhizal fungi, and other beneficial microorganisms that interact with the plant to enhance its defense mechanisms [306,307]. Certain strains of *Streptomyces* and *Bacillus* produce compounds capable of internally disrupting *Ganoderma* growth [204,308,309,310]. Additionally, systemic agents stimulate the tree’s defenses, aided by the symbiotic relationships with endophytic bacteria and mycorrhizal fungi [311,312]. Ongoing research is exploring advanced delivery methods like encapsulation and nanoformulation. Integrated management, combining biological control with cultural practices, offers a holistic approach for *Ganoderma* management in oil palm plantations.

*Streptomyces* GanoSA1 induces the production of pathogenesis-related proteins, including β-glucanase, peroxidase, polyphenol oxidase, and phenylalanine lyase, in oil palm seedlings, contributing to its effective biological control against *G. boninense* [310]. Encapsulated *Pseudomonas aeruginosa* showed strong bioactivity against *G. boninense*, inhibiting its growth by over 70%. Encapsulation improved biocontrol efficacy, making it a promising strategy for managing *Ganoderma* diseases in the oil palm industry [313]. Keratinolytic fungi A 12 and A 18 showed a strong antagonistic ability against *G. boninense*, inhibiting its growth and causing abnormalities in hyphae. These findings suggest their potential as biocontrol agents for agriculture [314]. Ligninolytic fungi from arrowroot, cocoyam, and canna plants inhibited *G. boninense* wood decay and reduced root infection in oil palm. They demonstrate potential for managing basal stem rot disease in oil palm plantations [315]. *Trichoderma asperellum* (UPM16) and *Bacillus cereus* (UPM15) effectively mitigated BSR disease in oil palm. The combination treatment of these biological control agents shows promise for sustainable BSR management in the oil palm industry [316].

A microbial community analysis of oil palm trees affected by basal stem rot identified differences between asymptomatic and symptomatic trees. Actinobacteria and Firmicutes were more abundant in the rhizosphere soil of asymptomatic trees, suggesting their potential as biological control agents against *G. boninense* [317]. Isolates AKT19, AKT28, and AKT52 of Streptomyces gelaticus demonstrated a strong suppression of *G. boninense* growth in dual culture tests. Volatile organic compounds and bioactive compounds from these isolates inhibited *G. boninense* growth by significant percentages. The isolates also produced enzymes and growth hormones. These findings highlight the potential of *Streptomyces gelaticus* as a biological control agent against *G. boninense* [318]. *Hendersonia toruloidea* (GanoEF1), a biocontrol agent, effectively suppressed BSR disease caused by *G. boninense* in oil palm seedlings [319]. Indigenous bacteria from peatlands, specifically three isolates suspected to be *Bacillus* species, exhibited inhibitory effects on the growth of *Ganoderma*, suggesting their potential as biocontrol agents [320]. Arbuscular mycorrhizal fungi (AMF) were found to promote the growth and resistance of oil palm seedlings against *Ganoderma* sp. The presence of *Ganoderma* sp. did not inhibit seedling growth, and no infection was observed in the roots. These findings highlight the potential of AMF in enhancing oil palm seedlings’ resilience to *Ganoderma* sp. infection [321]. Mycorrhizal application suppressed *Ganoderma* disease and promoted bole diameter growth in oil palm nurseries [322].

BSR disease in oil palm poses a challenge, and algae are explored as a potential source of natural compounds that can be used to inhibit *Ganoderma*. An in silico analysis identified serotonin, 5-methoxytryptamine, and melatonin as compounds with a strong binding affinity to target proteins. Further analysis and validation are needed to confirm their potential in inhibiting *Ganoderma* metabolism [323]. Bacterial isolates from peatlands show antagonistic activity against *Ganoderma* under low pH conditions, demonstrating potential as biological control agents for BSR disease in oil palm cultivation on peatlands [324]. Mycolytic enzyme-producing bacteria, including *Acinetobacter calcoaceticus*, *Chryseobacterium indologenes*, and *Pseudomonas putida*, exhibit antifungal properties against *G. boninense*. These bacteria have the potential to be effective biocontrol agents for managing basal stem rot disease in oil palm [325].

The soil microbiome influences BSR incidence in oil palm fields. A lower BSR incidence is associated with higher microbial diversities, favorable soil characteristics, and disease-suppressive bacteria. Managing the soil microbiome can help control BSR disease [326]. *Talaromyces apiculatus* (Cr) and *Clonostachys rosea* (Ta) biocontrol agents show potential in suppressing BSR and promoting plant growth in oil palm. They inhibit *G. boninense* growth and enhance leaf area, biomass, and nutrient content. The co-inoculation of Cr and Ta demonstrates improved disease control. These agents can be effective for BSR management while promoting plant growth [327]. Actinomycete strains, including *Streptomyces palmae* CMU-AB204T, show strong anti-*Ganoderma* activity and potential as biocontrol agents for basal stem rot (BSR) disease in oil palm. *Streptomyces palmae* CMU-AB204T reduces disease severity and enhances plant vigor. Bioactive compounds from *S. palmae* CMU-AB204T exhibit inhibitory activity against *G. boninense* [309]. *Streptomyces palmae* CMU-AB204T produces antimicrobial compounds effective against *G. boninense* and other bacteria and fungi. These compounds show potential for managing BSR disease in oil palm [328]. *Streptomyces nigrogriseolus* (GanoSA1) shows strong inhibition of *G. boninense* and positive traits as a potential biological control agent for *Ganoderma* disease in oil palm [329].

*Burkholderia* spp. strains recovered from oil palm plantations showed significant antagonistic activities against *G. boninense*. These strains, including *B. cepacia*, *B. contaminans*, *B. metallica*, and *B. stagnalis*, exhibit potential as biocontrol agents based on their antifungal properties and hydrolytic enzyme activities [330]. *Scytalidium parasiticum* shows potential as a biological control agent against *Ganoderma*. Co-cultivation induces the production of antimicrobial compounds, including eudistomin I, naringenin 7-O-beta-D-glucoside, penipanoid A, oleic acid, and stearamide. These compounds can help combat *Ganoderma* infections in oil palm fields [331]. BSR caused by *Ganoderma* is a major issue in oil palm plantations on peatlands. Biological control methods are being sought, and fungi from peatlands have shown antagonistic activity against *Ganoderma* at acidic pH levels, indicating their potential as control agents for BSR [332]. *Bacillus subtilis*, an endophytic bacterium, has shown potential as a biological resistance inducer against *G. boninense* in oil palm. It enhances growth, increases salicylic acid levels, and produces indole-3-acetic acid. *B. subtilis* holds promise as a natural control agent for *G. boninense* in oil palm seedlings [333]. AMF treatment enhanced enzyme activities and plant responses in *Ganoderma*-infected oil palm seedlings. Treated seedlings had a higher phenolic content and enzyme levels. AMF treatment improved plant growth and chlorophyll content, suggesting benefits for disease resistance in basal stem rot-affected plantations [334].

Seaweed extracts, particularly from *Caulerpa racemosa* var. lamouroxii, show strong antifungal activity against *G. boninense*. Phytol and l-(+)-ascorbic acid 2,6-dihexadecanoate have been identified as the dominant bioactive compounds. Malaysian seaweed species hold promise as a source of antifungal compounds for combating BSR disease in oil palm [335]. An endophytic bacteria isolate, identified as *Bacillus cereus*, demonstrated strong antifungal activity against *G. boninense*. This suggests the potential of oil palm plantation endophytic bacteria as a natural control method for *Ganoderma* infection [336]. Malaysian seaweed extracts inhibit *G. boninense*, offering potential antifungal compounds against basal stem rot in oil palm [337].

Alternative methods for controlling BSR in oil palm were investigated. *Streptomyces*-like actinomycetes isolates showed high similarity with known *Streptomyces* species. The most effective formulation, AGA347, reduced BSR in oil palm seedlings by 73.1%. Other isolates also demonstrated significant suppression of BSR. These findings highlight the potential of *Streptomyces* species as biological control agents against *Ganoderma* in oil palm [338]. Actinomycetes were isolated from soil samples in Sabah, Malaysia, to combat BSR caused by *G. boninense* in the oil palm industry. The *Streptomyces* strain A19 showed strong inhibition of *G. boninense* growth and caused damage to its hyphae. Metabolites from A19 exhibited potent inhibition, and fractionation enhanced their effectiveness. These findings highlight the potential of *Streptomyces* spp. A19 and its metabolites for controlling BSR in oil palm [339]. White-rot hymenomycetes isolated from oil palm trunks showed antagonistic potential against *G. boninense*. Oil palm seedlings inoculated with these hymenomycetes remained healthy and exhibited better growth compared to those infected with *G. boninense*. These findings suggest the potential use of white-rot hymenomycetes as biological control agents against *G. boninense* in oil palm plantations [340].

*Lentinus Cladopus* LC4 showed potential as a biological control agent against *G. boninense*, despite the absence of an inhibition zone. It could be used for preventing and controlling plant diseases, including *G. boninense* [341]. Endophytic bacteria were screened for their ability to control *G. boninense* in oil palm trees. Three promising strains (*Pseudomonas aeruginosa* GanoEB1, *Burkholderia cepacia* GanoEB2, and *Pseudomonas syringae* GanoEB3) inhibited *Ganoderma* growth in laboratory tests. Nursery trials showed reduced disease development in treated oil palm seedlings, with *P. aeruginosa* GanoEB1 being the most effective. Field studies are required to confirm their efficacy in controlling *Ganoderma* in oil palm plantations [342]. *Scytalidium parasiticum* showed strong inhibition against *G. boninense*, causing morphological changes in and the degradation of *Ganoderma* mycelia. Non-volatile metabolites of *S. parasiticum* suppressed *Ganoderma* growth. In nursery experiments, *S. parasiticum* reduced *Ganoderma* infection and disease severity while promoting oil palm growth. These findings highlight *S. parasiticum*’s potential as a biocontrol agent against BSR in oil palm [343,344]. Endophytic bacteria were screened as biological control agents against *G. boninense* in oil palm. *Pseudomonas aeruginosa* GanoEB1, *Burkholderia cepacia* GanoEB2, and *Pseudomonas syringae* GanoEB3 inhibited the pathogen’s growth. Nursery trials showed reduced disease incidence and severity in treated seedlings, with *P. aeruginosa* GanoEB1 being the most effective. Field studies are needed to confirm their efficacy [342].

The effectiveness of combining arbuscular mycorrhizal fungi (AMF) with endophytic bacteria (EB) in reducing BSR disease in oil palm was investigated. Nursery and field trials using pre-inoculated seedlings demonstrated a significant reduction in disease development. The combination of AMF and EB showed improved biocontrol efficacy, highlighting its potential for controlling BSR in oil palm plantations [345]. Isolated actinomycetes from the oil palm rhizosphere showed a significant inhibition of *G. boninense*. Four specific isolates (AGA 043, AGA 048, AGA 347, and AGA 506) demonstrated strong inhibitory effects and potential metabolites against *G. boninense*. These findings highlight the potential of these actinomycetes as biocontrol agents for managing BSR [346]. *Burkholderia* GanoEB2 in formulated bioorganic powders effectively suppressed *G. boninense* in oil palm seedlings. Real strong bioorganic fertilizer (RSBF) achieved a higher disease reduction (85.3%) compared to bioorganic empty fruit bunch (BEFB) (70.5%) [347].

*Cladobotryum semicirculare* demonstrated potent antagonistic activity against *G. boninense* and other fungal pathogens. It inhibited mycelial growth and displayed mycoparasitic behavior, reducing the re-isolation of *G. boninense* and *G. lucidum* mycelia. *Cladobotryum semicirculare* was identified as a host for *G. boninense* [348]. Chitinolytic bacteria (*Bacillus amyloliquefaciens* and *Serratia marcescens*) showed a high inhibition of *G. boninense* growth. They exhibited optimal chitinase production at different incubation times, and their chitinase activity was effective over wide pH and temperature ranges. The chitinase from these bacteria successfully inhibited *G. boninense* mycelium growth [308]. Combining arbuscular mycorrhizal fungi (AMF) with endophytic bacteria (EB) effectively reduced BSR disease in oil palm. Nursery and field trials using pre-inoculated seedlings showed significant disease reduction. The combined treatment demonstrated improved biocontrol efficacy, indicating its potential for controlling BSR in oil palm plantations. Bait seedlings were valuable for studying BSR development [345].

*Gliocladium* spp. have shown promise in controlling *Ganoderma* basal stem rot disease in oil palm [349]. *Burkholderia* sp. isolated from symptomless oil palm showed potential as a biological control agent against *G. boninense* in vitro. Antifungal genes were detected in *Burkholderia* sp. However, in vivo tests on oil palm seedlings did not reduce disease incidence or severity. Further research is needed to evaluate its efficacy under field conditions [350].

Combining *Trichoderma* and vesicular arbuscular mycorrhizae (VAM) shows promise in controlling BSR in oil palm [351]. Chitinolytic bacterial isolates inhibited *G. boninense* growth and induced hyphal abnormalities. Applied to oil palm seedlings, they reduced basal stem rot disease incidence. These isolates show potential as endophytes for suppressing *G. boninense* in oil palm [352]. *Burkholderia* GanoEB2 powder has been found to be effective in controlling and suppressing *Ganoderma* under laboratory conditions and in nursery trials [353]. Endophytic bacteria from oil palm roots, *Pseudomonas aeruginosa* UPMP3 and *Burkholderia cepacia* UMPB3, enhance the growth of arbuscular mycorrhizal fungi (AMF) and exhibit antagonistic effects against *G. boninense* [309]. Six hundred actinomycete isolates were obtained from soil samples in *Ganoderma*-affected oil palm plantations. Among them, 568 were identified as Streptomyces, and 32 were non-Streptomyces. In vitro tests revealed that 81 isolates displayed strong antifungal activity against *G. boninense* [354].

Endophytic bacteria isolated from healthy oil palm roots showed inhibitory effects against *G. boninense*. Four bacteria (EB2, EB4, EB5, and EB6) demonstrated potential as biocontrol agents, with *Pseudomonas aeruginosa* (EB6) being particularly effective. These findings suggest their potential for controlling BSR disease [355]. The *Burkholderia* strain UPM B3, isolated from oil palm roots, was identified as a non-pathogenic species within the *Burkholderia cepacia* complex. It exhibited antagonistic activity against *G. boninense*. This suggests that UPM B3 has the potential to be used as a biocontrol agent against BSR [356].

The endophytic bacteria *Burkholderia cepacia* (B3) and *Pseudomonas aeruginosa* (P3) have potential as biocontrol agents against BSR in oil palm. They invade and proliferate within the plants, suppressing *G. boninense* and improving seedling growth [357]. *Ganoderma* can degrade lignin and utilize cellulose. Understanding *Ganoderma* as a white-rot fungus is crucial for controlling it in oil palm. This perspective offers opportunities for breeding resistant cultivars with a high lignin content and implementing practices to prevent decay. Spore dispersal is a likely mode of spread. Applying this knowledge can help efficiently degrade oil palm waste. Considering the white-rot process is essential for effective *Ganoderma* control in oil palm [358]. Molecular and immunological methods were employed to detect *Ganoderma* disease in coconut plants. Effective treatments included the combination of *Pseudomonas fluorescens* and *Trichoderma viride* with chitin amendments, as well as integrated disease management and fungicide tridemorph treatments. *Trichoderma harzianum* and *P. fluorescens* + *T. viride* treatments showed lower infection levels [359].

Arbuscular mycorrhizas (AMs) enhanced oil palm seedling resistance to *G. boninense*. AM inoculation increased root density, improved nutrient status, and delayed disease symptoms. Mycorrhiza-treated seedlings had longer roots and higher lignin and calcium levels, enhancing pathogen resistance [360]. AM inoculation improved oil palm seedlings’ root density, nutrient status, and resistance to *G. boninense*. Treated plants showed longer roots, reduced disease severity, and higher lignin and calcium contents. AM has potential as a biological control strategy in oil palm cultivation [360]. AM fungi enhance oil palm seedling growth and act as a biocontrol agent against *G. boninense*-induced basal stem rot (BSR). The optimal inoculum density of 40 g/plant improves growth and colonization. AM association improves physiology and root morphology. Inoculated seedlings show delayed BSR symptoms and reduced mortality. AM fungi show potential as a biocontrol agent, promoting growth and reducing BSR severity in oil palm [361].

#### 7.3.3. Other Methods

Naturally occurring phenolic compounds have been evaluated as a sustainable method to suppress the pathogens and their impact on the cell membrane potential. This approach aims to reduce reliance on synthetic fungicides and minimize environmental impact [362]. *Hendersonia* GanoEF biofertilizer effectively controls *Ganoderma* disease in oil palm seedlings, resulting in improved seedling growth and reduced disease incidence and severity [363]. GanoCare^®^, a new fertilizer technology, shows efficacy in suppressing BSR caused by *G. boninense* in oil palm. It improves growth parameters and reduces BSR incidence and severity in oil palm seedlings. The most effective treatment involves pretreatment and continuous application every three months. GanoCare^®^ enhances the seedlings’ defense system against BSR [364]. Crude phenazine synthesized by *Pseudomonas aeruginosa* UPMP3 shows promise as a biocontrol agent against BSR in oil palm. Its application significantly reduces disease severity and improves plant vigor compared to the fungicide hexaconazole. The upregulation of specific genes suggests the involvement of induced resistance in controlling the pathogenic *G. boninense* [365].

The effectiveness of the Colonized System of *Ganoderma* Vaccine (CHIPS^®^) as a biocontrol agent against *Ganoderma* disease in oil palm trees was evaluated. CHIPS^®^ treatment showed a significant decrease in the disease severity index (DSI) compared to other treatments, offering a higher profit ratio. These findings have practical implications for managing *Ganoderma* disease in oil palm cultivation [366]. The effectiveness of biological control agents (BCAs) in managing *G. boninense* was investigated. BCA treatments were tested in nursery and field settings, assessing disease progression and analyzing *Ganoderma* biomass. The BCAs did not prevent BSR infection in nursery trials but disrupted the pathogen’s progression based on ergosterol content [367].

Naturally occurring phenolic compounds were evaluated for their inhibitory effects on *G. boninense* enzymes. Benzoic acid demonstrated the highest effectiveness. These compounds show promise as alternatives to chemical inhibitors for controlling *Ganoderma* infections [368]. The CuSO_4_ treatment of oil palm seedling roots increased phytochemical content and enhanced antifungal activity against *G. boninense* [369]. Effective methods for controlling *G. boninense* were investigated. Plant extracts, including *Antigonon leptopus* and *Carica papaya* leaves, *Zingiber montanum* and *Curcuma longa* rhizomes, and *Carica papaya* latex, showed limited effectiveness. *Carica papaya* extract exhibited a partial inhibition (41.26%) of mycelium growth. Indigenous antagonistic microorganisms, such as bacterial isolates B001, B002, and B003, and fungal isolate T0 demonstrated high efficiency in controlling the pathogen [370].

The Indonesian Research Institute for Biotechnology and Bioindustry developed organic fungicide formulas to control *Ganoderma* sp. in oil palm cultivation. Weekly application of the fungicide promoted palm root growth, improved plant performance, induced flower formation, and increased nutrient levels. Fruit production and oil content were higher with organic fungicide treatments, particularly with applications every two weeks [371]. Papaya leaf extracts exhibit antifungal properties against *G. boninense*. Methanol and acetone extracts are effective in inhibiting fungal growth, with higher concentrations showing stronger effects. The extracts contain bioactive compounds that have the potential to hinder the growth of *G. boninense* [372]. Three biological control agent (BCA) products successfully reduced *G. boninense* colonization in oil palm trials, lowering ergosterol content and disease incidence (DI). TR1 and TR3 significantly reduced DI to 12% and 24%, respectively, while decreasing ergosterol levels in trunk tissues. BCAs hold promise for *Ganoderma* control in oil palm [373].

ABM-OP1, an enzyme-based biological product developed using the ABM (Advanced Beneficial Microbialsystem), effectively destroys over 85% of internal *Ganoderma* mycelia in oil palm. Spraying on trunk cross-sections and targeted injection strategies achieve a high efficacy of 80% to 96% in eliminating internal mycelia. ABM-OP1 shows promise for controlling *Ganoderma* infection and prolonging oil palm tree productivity [374].

### 7.4. Chemical Control

Dazomet–micelle nanodelivery systems (DAMINs) using sodium dodecylbenzene sulfonate (SDBS) as a surfactant showed the highest inhibitory activity against *G. boninense*. These DAMINs have the potential to be effective nanofungicides for disease control [375]. *The ERG11* gene in *G. boninense* involved in ergosterol biosynthesis and targeted by fungicides was isolated and characterized. Its upregulation during interaction with oil palm provides valuable insights for developing control measures against BSR in oil palm plantations [193]. Anionic surfactant-based nanodelivery systems demonstrate stronger antifungal effects against *G. boninense*, showing potential as effective agronanofungicides for managing basal stem rot disease in oil palm [376].

Different phenolic compounds were tested for controlling *G. boninense*. Among them, benzoic acid treatment at 1 mM resulted in only 31% wood mass loss after 120 days, indicating its potential as a sustainable solution for reducing *G. boninense* inoculum pressure during replantation in the oil palm industry [377]. Fungicide nanodelivery agents based on hexaconazole–micelle systems show potential for combating basal stem rot disease caused by *G. boninense* in the oil palm industry. Tween 80, a non-ionic surfactant, exhibits the lowest effective concentration (EC50 value of 2), indicating improved efficacy against the fungi. The findings suggest the use of agronanofungicide for treating *Ganoderma* disease in oil palm plantations [378]. The impact of commonly used herbicides on *Ganoderma* disease in oil palm plantations was investigated, and in vitro experiments revealed paraquat dichloride as the most effective in inhibiting *Ganoderma* growth. Nursery trials showed that diuron-treated seedlings had the highest disease progression [379]. Benzoic acid treatment shows potential in suppressing *G. boninense* and preventing wood decay in oil palm trees affected by BSR. It inhibits fungal growth, causes cellular damage, and reduces wood weight loss. This suggests benzoic acid as a viable management option for BSR in oil palm production [380].

Chitosan nanoparticles with hexaconazole and/or dazomet effectively combat BSR in oil palm caused by *G. boninense*. They act as biocides and nanocarriers, with their size influencing disease suppression; 5 nm chitosan–hexaconazole–dazomet nanoparticles show the highest disease reduction, offering eco-friendly control for basal stem rot [381]. A chitosan–hexaconazole nanoformulation effectively combats *G. boninense* in oil palm trees, with no fungicide residues in palm oil. It accumulates in the stem tissue and leaves, exhibiting long-lasting effects with half-lives of 383 and 515 days [382]. Tetramethylthiuram disulfide (thiram) exhibits antifungal properties against *G. boninense*, with higher concentrations completely inhibiting its growth. An ergosterol analysis confirmed the effectiveness of thiram treatment, showing a lack of detectable ergosterol in completely inhibited *G. boninense* [383].

Chitosan nanoparticles were developed as a nanodelivery system for hexaconazole to control *G. boninense* in oil palm. The nanoparticles exhibited sustained release of the fungicide, and smaller particle sizes showed higher antifungal activity. This system holds potential for effective fungicide delivery and *G. boninense* control [384]. Researchers studied lignin-degrading enzymes in *Ganoderma* species, specifically *G. boninense* (PER71). *Ganoderma boninense* showed high enzyme activity and effective decolorization. Enzyme inhibitors, including EDTA and thioglycolic acid, effectively reduced enzyme production. These findings have implications for the chemical control of basal stem rot [385]. Pyraclostrobin effectively suppresses *G. boninense*. In vitro and in vivo experiments demonstrated its inhibitory effects, reduced infection, and improved plant growth. Pyraclostrobin shows promise as a treatment for basal stem rot in oil palm [386]. *Ganoderma boninense* thrives on fructose and glucose as carbon sources and on ammonium citrate and ammonium nitrate as nitrogen sources. It prefers a pH of 4–5, a humidity of 50–60%, and temperatures between 25 and 32 °C. These insights can aid in controlling *Ganoderma* disease in oil palm plantations [387]. Researchers investigated the biochemical interaction between *G. boninense* and oil palm seedlings using a metabolomics approach. They identified asparagine and chelidonic acid as potential markers for distinguishing different treatment types in *Ganoderma* infection in oil palm seedlings [388].

Hexaconazole was successfully intercalated into zinc/aluminum-layered double hydroxide (ZALDH) using an ion-exchange method, resulting in improved thermal stability. The nanocomposite demonstrated enhanced inhibition against *G. boninense* growth compared to free hexaconazole. This research offers potential for combating basal stem rot [389]. Phenolic acids were found to inhibit the ligninolytic enzyme activity of *G. boninense*. Syringic and caffeic acid showed significant inhibitory effects. These findings highlight the potential of phenolic acids as effective inhibitors for controlling *Ganoderma* infections in oil palm plantations [390]. Benzoic acid and salicylic acid were evaluated for their effectiveness in controlling BSR in oil palm. The exogenous application of these phenolic compounds suppressed BSR and promoted plant growth by increasing defense enzymes and lignin content. Benzoic acid at 10–15 mM effectively reduced BSR, while salicylic acid showed mixed results. Benzoic acid at a 15 mM concentration could be a natural control agent for BSR in oil palm [391]. Salicylic acid inhibited *G. boninense* growth, while jasmonic acid promoted it, revealing isolate-specific effects and potential as fungicides [392].

The effectiveness of prochloraz, kresoxim methyl, and chlorotalonil in controlling *G. boninense* was assessed through in vitro screening. These fungicides demonstrated high efficacy, with respective percentages of mycelium growth inhibition of 96.22%, 98.96%, and 88.44% [370]. Plant phenolic compounds were tested as controls against *Ganoderma*. Most compounds inhibited *Ganoderma* growth and reduced wood-degrading enzyme production. Benzoic acid showed the strongest inhibitory effect. Hence, these findings suggest the use of phenolic compounds for the potential control of basal stem rot in oil palm [393]. Mineral nutrient and salicylic acid (SA) supplementation effectively controlled BSR disease in oil palm seedlings. Treatment with calcium/copper/SA showed the lowest disease severity (5.0%) and reduced fungal activity. It promoted lignification, delayed BSR symptoms, and suppressed the disease while promoting seedling growth [394]. Optimum nitrogen (N), phosphorus (P), and potassium (K) concentrations were identified to enhance oil palm seedlings’ resistance against *Ganoderma* disease. Specific concentrations in a solution culture and peat soil promoted better growth and defense mechanisms against *Ganoderma*, aiding in reducing incidence and severity in oil palm plantations [395].

In oil palm seedlings, Si treatment reduced BSR severity caused by *G. boninense*. SiO_2_ at 1200 mg L^−1^ showed the highest disease suppression, with a 53% reduction in incidence. Si treatment restricted fungal penetration and improved resistance, demonstrating potential economic benefits [396]. A combination of phenolic acids effectively suppressed *G. boninense* in oil palm trees affected by BSR disease. The phenolic acid treatment reduced *Ganoderma* colonization in palm tissues, showing potential for controlling BSR [397]. The GanoCare™ OC Special can be used as a preventive treatment to control *Ganoderma* infection in oil palm, reducing potential yield losses in oil palm cultivation [398]. Hexaconazole, a fungicide used to control BSR disease in oil palm, was studied for its dissipation rate in soil. Residues were detected in soil samples, with a decreasing dissipation rate over time. The half-life of hexaconazole ranged from approximately 69.3 days to 86.6 days, depending on the dosage. Soil conditions, environmental factors, and the chemical properties of hexaconazole influenced its dissipation in humid soil [399].

Phenolic acids were tested for suppressing BSR disease in oil palm. Different concentrations of these acids were applied to trees with a similar BSR intensity. Treated palms had lower ergosterol levels compared to untreated infected palms. The combination of phenolic acids at 1.6 g showed the highest suppression of *Ganoderma* colonization. A number of *G. boninense* colonies were found in soil samples after one month [400]. Cu and Ca supplementation in oil palm significantly reduced *G. boninense* infection, leading to slower disease progression and delayed onset. Combined Ca and Cu treatment resulted in longer symptom-free periods. The supplementation enhanced oil palm’s resistance mechanisms, promoting the production of peroxidase and lignin during fungal penetration [401]. The fumigant methylisothiocyanate (MITC) released by dazomet has the potential to restrict *Ganoderma* infection spread in standing palms, extending their lifespan. Dazomet applications improve oil palm productivity by eliminating *Ganoderma* inoculum spread within infected palms [402].

Chitosan effectively suppressed *Ganoderma* infection in oil palm seedlings at a minimum concentration of 0.5%. It reduced disease severity and hindered fungal sterol production. Bole tissue infection was not significant, but *Ganoderma* was isolated from the roots in all treatments, except for the healthy control [403]. Ca^2+^, Cu^2+^, and SA together effectively controlled *G. boninense* in vitro. SA alone also showed effectiveness. Pretreating *Ganoderma*-infected rubber wood blocks with the combination reduced basidiocarp size, weight, and wood block weight loss. The combination demonstrates potential for suppressing *G. boninense* growth in vitro [404]. Chitosan controlled *Ganoderma* infection in oil palm nurseries. A 0.5% concentration effectively reduced disease severity and fungal sterol levels. Tissue infection did not vary significantly among chitosan concentrations. Pretreatment with chitosan showed slightly lower disease severity compared to post-infection treatment [405]. Phenolic acids in oil palm roots interact with *G. boninense*. Syringic acid, caffeic acid, and 4-hydroxybenzoic acid were examined. An ergosterol analysis showed a correlation with BSR severity. Chitosan stimulates phenolic acid production. Phenolic acids have potential for managing BSR disease in oil palm [406].

Oil palm disease resistance against *G. boninense* was studied. Phenolic compounds and chitosan were examined. AVROS had higher phenolic levels. Chitosan boosted phenolic accumulation, but it decreased in infected tissues. Syringic acid showed strong antifungal activity. Phenolic acid accumulation, especially syringic acid, could be a marker for breeding resistant oil palm. More research is needed for BSR management [407]. Salicylic acid can effectively suppress *Ganoderma* spp. in oil palm seedlings. Concentrations of 50 and 100 ppm inhibited *Ganoderma* growth by 100%. Higher concentrations (100–200 ppm) improved seedling growth and chlorophyll content. Salicylic acid shows potential for suppressing *Ganoderma* and enhancing oil palm seedling growth [408]. The phenolics present in oil palm root, including syringic acid, caffeic acid, and 4-hydroxybenzoic acid, have fungitoxic properties. Syringic acid effectively inhibits the growth of *G. boninense*, even at a low concentration of 0.5 mg/mL. Caffeic acid and 4-hydroxybenzoic acid also suppress the growth of *G. boninense* at higher concentrations [409]. The synergistic effect of three phenolic compounds (syringic acid, caffeic acid, and 4-hydroxybenzoic acid) from oil palm roots was investigated against *G. boninense*. Combinations of these phenolic compounds at concentrations ranging from 0.5 to 2.5 mg/mL exhibited strong fungitoxic effects on *G. boninense*. The combination of 4-hydroxybenzoic acid, syringic acid, and caffeic acid showed a significant inhibition of *G. boninense* growth [410]. White rot refers to fungi attacking lignin in woody tissue, exposing cellulose. By modifying oil palm lignin, losses from *Ganoderma* infection could be reduced [411]. Controlling BSR in oil palm involves inhibiting enzymes of *Ganoderma*, but specific examples of enzyme inhibition are unknown. Factors like temperature, pH, and the carbon-to-nitrogen ratio can potentially inhibit these enzymes. In vitro testing is needed to develop control methods for oil palm and other wood rot diseases. Inhibitors can be applied via injection, spraying, or soil treatment [412].

Calcium nitrate was studied as a preventive measure against BSR caused by *Ganoderma* in oil palm seedlings. Applying calcium nitrate one month before inoculation significantly suppressed BSR incidence, while soil augmentation with *Trichoderma* after inoculation did not show significant control. The researchers suggest that calcium strengthens seedling cell walls and promotes the growth of antagonistic fungi, potentially reducing BSR [413]. *Ganoderma* spp. isolated from infected arecanut palms in Karnataka, India, were found to cause disease in healthy palms. The severity of the disease varied depending on the method of inoculation. Digging a trench around affected plants reduced disease spread, and spraying with captan or Bavistin (carbendazim) prevented its spread [414]. Four chemicals (tridemorph, peconazole, triadimenol, and quintozene) were tested for their fungitoxicity against three fungal pathogens. All chemicals showed high effectiveness, with quintozene being relatively less toxic. Peconazole exhibited excellent vapor phase inhibition, while triadimenol had minimal vapor phase activity. Tridemorph and quintozene remained effective for up to 8 weeks after soil treatment [415]. With *Ganoderma* infection in oil palms, the fungicides triademefon, triadimenol, methfuroxam, carboxin, biloxazol, and cycloheximide have been found to be effective against *Ganoderma* sp. [416].

### 7.5. Host Resistance

Mixed cropping with water yam reduces *G. boninense* infection severity in oil palm, resulting in reduced root necrosis, lower plant mortality, and decreased pathogen inoculum potential. Fungal survival is suppressed, and oil palm seedling growth remains unaffected. Mixed cropping shows promise as a strategy to control *G. boninense* damage in oil palm plantations [417]. Molecular studies revealed the involvement of protein-coding genes and non-coding RNAs in the interaction between *G. boninense* and oil palm. A computational analysis predicted a diverse set of non-coding RNAs in both genomes, providing insights into the molecular mechanisms of infection and potential strategies for disease management [418].

Marker-assisted selection (MAS) identifies genetic markers for *Ganoderma* resistance in oil palm breeding, facilitating the selection of resistant planting material with reduced field infection rates [419]. The genetic diversity and population structure of *G. boninense* were examined using cDNA-SSR markers. High variability and gene flow were observed, indicating regional adaptation and multiple genetic sources. This research provides insights for effective BSR disease control [197]. Resistant oil palm plants showed higher yields and lower *Ganoderma* infection rates compared to non-resistant plants over 25 years. Resistant plants yielded 581,163 fresh fruit bunches per hectare (FFB/ha), while non-resistant plants yielded 385,158 FFB/ha. *Ganoderma* disease incidence was significantly lower in resistant plants (19.95%) compared to in non-resistant plants (86.42%). Survival rates were higher for resistant plants throughout the 25-year period [420].

Genetic engineering was used to develop fungus-resistant oil palm plants by introducing the *AGLU1* and *RCH10* genes, along with a Basta^®^ resistance gene. Among the regenerated plants, two demonstrated resistances to *G. boninense* fungus, and this showcases the potential of genetic engineering in creating disease-resistant oil palm varieties [421]. Developing *Ganoderma*-resistant oil palm planting material is crucial for managing basal stem rot disease. The Indonesian Oil Palm Research Institute has made progress in identifying genetic resources, conducting a pedigree analysis, and testing resistant materials. The promising results indicate the presence of *Ganoderma*-resistant planting materials, providing potential solutions for disease control [422].

The genetic variation of *G. boninense* was investigated in oil palm plantations in Sarawak, Malaysia. Dominant haplotypes were found across all locations, indicating a single founder population adapted to oil palm. However, it is important to note that this does not necessarily imply a racial structure of the pathogen according to a gene-by-gene pattern. Understanding these genetic characteristics is crucial for managing BSR in oil palm plantations [7]. A new mycelium inoculation technique was introduced to assess oil palm resistance to BSR caused by *G boninense*. This faster and simpler method allows for consistent infection, early disease evaluation, and differentiation between resistant and susceptible palm seedlings [423].

Genetic loci associated with *Ganoderma* resistance in oil palm were identified through a long-term study using a multi-parental population. This provides valuable information for breeding programs to develop resistant oil palm varieties, combating the damaging pathogen [34]. *Ganoderma* infection in oil palm trees is influenced by soil type, with acid sulfate and potential acid sulfate soils leading to a lower life expectancy. Palms showing foliar symptoms have shorter survival durations. Improved management strategies are required, especially for oil palms planted in coastal soils with acid sulfate layers [424]. Differentially expressed genes associated with *Ganoderma* tolerance in oil palm plants were identified. Six potential positive biomarkers for *Ganoderma* tolerance were discovered among the 21 genes tested [425].

Partial resistance to basal stem rot caused by *G. boninense* has been observed in oil palm. A methodology has been developed to screen and select oil palm varieties with high resistance. Through an evaluation of thousands of crosses, Deli and la Mé origins with partial resistance have been identified. These findings have led to the development of a new oil palm variety with improved resistance to basal stem rot [426]. *Polyalthia longifolia* (glodokan) showed greater resistance to *Ganoderma* sp. infection compared to *Pterocarpus indicus* (angsana), making it a more suitable tree species for planting in affected areas. The findings provided guidance for selecting trees at Gadjah Mada University (UGM) campus [427].

No effective control method or resistant oil palm variety has been found for managing BSR. AVROS, a commonly planted variety, is believed to be more resistant, but a study found that the Calabar and Ekona varieties were actually more susceptible based on their higher ergosterol content and disease severity scores. Ergosterol accumulation increased over time in all three varieties [428]. At the MPOB Research Station in Malaysia, oil palm dura progenies were evaluated for their susceptibility to BSR. The selfed progenies were more susceptible than the inter-crossed ones. Progeny MS3358 with an Elmina origin showed greater tolerance to the disease, indicating its potential for developing *Ganoderma*-tolerant planting materials [429].

Breeding trials aim to develop *Ganoderma*-tolerant/-resistant planting material for oil palm. Genetic variation allows for the production of tolerant oil palm generations. Identifying alternative breeding material and exploring additional prevention measures are crucial for effective *Ganoderma* disease management [430]. A screening test was developed for oil palm progenies to detect resistance to BSR. It showed reliable early selection and consistent results, and it prevented the planting of highly susceptible progenies. The test is efficient, assessing around 100 progenies per month [431].

BSR is a severe disease in oil palm. Cultural practices and biological control have been used, but a genetic approach is now being considered. Early nursery screening tests are crucial, using parameters like *G. boninense* aggressiveness and the incubation period. Optimized parameters save time and space, enabling the selection of resistant varieties. Preliminary results show consistent susceptibility levels, but field observations are needed for confirmation [432]. Field observations in North Sumatra, Indonesia, showed varying susceptibility to basal stem rot caused by *G. boninense* in oil palm estates. The Deli origin material of *Elaeis guineensis* was highly susceptible, while African origin material was less susceptible. Variations in disease reaction were observed within origins and clones. Early selection tests are crucial for disease management [433]. Oil palm progenies were evaluated for resistance to *G. boninense*. All tested progenies were susceptible, showing a varying severity of foliar symptoms. PK 2724 (DxD) was the most susceptible, while PK 2567 (DxP) displayed partial resistance. Developing resistant progenies is considered a long-term solution for managing BSR disease [62].

### 7.6. Soil Management

The phosphorus levels in BSR-infested oil palm plantation soil may contribute to susceptibility to *G. boninense*. Effective soil management is crucial for disease control. A study compared BSR-suppressive and BSR-conducive soils, finding that sterilizing the suppressive soil increased BSR severity, while the non-sterilized suppressive soil had lower levels of the BSR-causing fungus [434]. Peat soil promotes faster and more severe disease progression compared to mineral soils. Organic matter from soil and oil palm residues contribute to the disease cycle [435].

Soil pH was studied to address BSR disease in oil palm caused by *G. boninense*. Different pH levels were tested, and pH 6 was found to effectively suppress BSR development in oil palm seedlings [436]. Peat land characteristics and biological factors were examined to understand the intensity of *Ganoderma* attacks in oil palm cultivation. Groundwater level and pH showed a significant correlation with *Ganoderma* attacks, highlighting the importance of soil management practices in controlling the disease [332]. BSR disease in oil palm revealed higher *Ganoderma* infection in replanting areas. BSR incidence decreased with surgery-mounding and *Ganoderma* removal during replanting. Proper debris removal and planting methods are crucial to minimize BSR and early-stage *Ganoderma* infection in oil palm crops [437]. *Ganoderma* species with chlamydospores showed better survival rates at various soil moisture levels. Flooding was effective in controlling species without chlamydospores. Woody debris supported long-term survival, and chlamydospores enhanced resistance to flooding [438].

### 7.7. Integrated Disease Management (IDM)

*Ganoderma boninense* inhibition in oil palm was influenced by high temperatures and mancozeb fungicide, which reduced enzyme activities and increased potassium levels, while boron inhibited specific enzymes. *Trichoderma virens* K1-02 showed antagonistic effects through antifungal compounds and enzymes, supporting integrated disease management for basal stem rot [3]. The *Ganoderma* vaccine/Biofungicide CHIPS and MOAF fertilizer combination effectively managed BSR in oil palms. It outperformed inorganic fungicides and *Trichoderma* treatments, resulting in high productivity. This integrated approach shows promise for BSR management in oil palm plantations, even in unfavorable environments [439]. *Trichoderma* and mycorrhizae treatments improved root health and crown dry weight in oil palm seeds, while KCl fertilizer primarily influenced root health. The recommended treatment combination is T1M2K1 for effective *Ganoderma* disease management [440].

Nursery screening is a convenient and rapid method for evaluating biopesticide effectiveness. Key parameters, including shading, wood block quality, inoculation stage, and environmental conditions, should be standardized. The step-by-step process involves experimental design, nursery preparation, the selection of *Ganoderma*-inoculated wood blocks, and the *Trichoderma* inoculation of seeds/seedlings [441]. BSR caused by *Ganoderma* is a destructive disease in palms. Early diagnosis is challenging, and symptoms appear late. Effective control involves calcium nitrate soil application and a combination of fungicides and cultural practices to manage BSR and improve disease management in palms [442].

## 8. Other Crops Reported with Basal Stem Rot

In Kerala, India, an outbreak of stem rot in coconut palms caused by *Ganoderma* species raised concerns among farmers. Previous studies had associated various *Ganoderma* species, including *G. applanatum*, *G. boninense*, *G. lucidum*, *and G. zonatum*, with this disease. Two new species, *Ganoderma keralense* and *G. pseudoapplanatum*, were proposed, highlighting their unique genetics and geographic distribution patterns [443]. *Ganoderma* wilt (BSR) caused by *G. boninense* affects coconut production in Southern Karnataka, India. A survey found higher incidences in Tumkur and in coconut–arecanut intercropping. Factors like sandy and red soils, canal water, and flood irrigation contribute to increased disease rates. Regularly cultivated gardens, particularly those in the 30–50-year age group, show higher susceptibility [444].

*Ganoderma boninense* was identified as the cause of BSR in foxtail palm trees. Symptoms included wilted leaflets, collapsed fronds, and necrotic basal stem bark. The fungus was confirmed as pathogenic through isolation, microscopic observation, and molecular identification. This discovery highlights the threat posed by *G. boninense* to the survival of foxtail palm trees and urban landscapes [445]. Arecanut cultivation in Assam, India, faces challenges, including basal stem rot caused by *G. lucidum*. The disease has a long history and varying incidence across districts. Neglected, poorly drained, and overcrowded gardens experience more severe cases [446]. BSR caused by *Ganoderma* is a significant problem in arecanut production in South Karnataka, India. Disease incidence varies from 0 to 55% in the region, with Tumkur having the highest incidence (19.3%). Sandy soils (13.5%) and canal irrigated gardens (12.7%) show a higher incidence. The pathogen was isolated from *Ganoderma* sporophores and infected root bits. Pot culture tests revealed the virulence of different *Ganoderma* isolates, with AG 9 being the most severe (75 DSI) [447].

In southern Karnataka, India, *Ganoderma* wilt disease was studied in coconut plantations. Tumkur district had the highest incidence (9.09%), followed by Hassan and Chitradurga districts. Disease incidence varied within districts and among specific talukas and coconut gardens. The disease was observed in different cropping patterns, including sole coconut crops [448]. *Ganoderma lucidum* caused significant mortality in Neem trees at the University of Port Harcourt in Nigeria. Over 40% of the 35-year-old trees were affected. The fungus was the most prevalent during the rainy season, damaging roots and stem butts. Older trees, soil contact, and moisture influenced the severity. The selective harvesting of affected trees and replacement with non-vulnerable species are recommended [449]. Etiology and epidemiological studies on the *Ganoderma* wilt of coconut trees identified *G. lucidum* and *G. applanatum* in 37 out of 86 samples. Sporophores and diseased root bits were effective sources for isolating *Ganoderma*. Basal stem rot incidence varied with rainfall, showing lower rates in areas with heavy rainfall. Rainfall negatively influenced vertical spread but had no clear relationship with horizontal spread [450].

Basel stem rot is a lethal and incurable disease affecting mature date palms and is caused by *G. zonatum* (MURRILL), *G. boninense*, *and G. tornatum* [451]. *Ganoderma applanatum* and *G. lucidum* Karst are fungal species that cause stem, butt, and root rot diseases in various tree species in Pakistan, including in *Pinus*, *Dalbergia*, *Artocarpus*, *Morus*, *Cedrus*, *Melia*, *Quercus*, and *Populus* [452]. A genetic analysis revealed variability and a close relationship between *G. boninense* isolates from oil palm and *Ganoderma* species from coconut stumps in Malaysia. This finding has implications for disease control and replanting strategies [453]. One study provides an overview of basal stem rot symptoms in Sri Lankan coconut and betelnut palms. It also investigates the molecular variation and somatic incompatibility groupings of *Ganoderma* isolates from Sri Lankan coconuts using mitochondrial DNA and amplified fragment length polymorphism profiles. The variation is compared to isolates from Malaysian coconuts, where *Ganoderma* is not known to be a pathogen [454].

A major outbreak of *Ganoderma* rot and bole rot caused by *G. boninense* occurred in a coconut estate in Ambalantota, southern Sri Lanka. Around 350 coconut palms were infected, resulting in fatal root and bole rot [455]. Coconut palms in Sri Lanka are affected by a tapering disease caused by *G. boninense*. Symptoms include pinnae necrosis, leaf drop, the necrosis of male flowers, and reduced nut setting. Extensive root decay, bud soft rot, stem bleeding, and bole rotting are observed. The fruit bodies of the fungus appear at the base of the bole [456]. *Ganoderma boninense* causes coconut tree disease in Sri Lanka. Symptoms include leaflet necrosis, leaf shedding, male flower necrosis, and reduced nut production. The disease is accompanied by extensive root decay, bud soft rot, stem bleeding, and trunk rotting in advanced stages. The fruit bodies of the pathogen are commonly found at the base of the trunk [456]. *Ganoderma boninense* causes tapering disease in mature coconut plantings, affecting palms over 40 years old. Symptoms include the yellowing and drying of lower fronds, smaller fronds and nuts, and a tapering trunk leading to crown collapse. Precautions should be taken to prevent infection in young plants and ensure the safety of replanted areas [457]. A summary of the control methods for BSR in oil palm is presented in Table 2.

## 9. Research on Detection Methods for Basal Stem Rot of Other Crops

The detection of *Ganoderma* disease in coconuts was performed using molecular and immunological methods. Field trials with various treatments, including *Pseudomonas fluorescens* + *Trichoderma viride* and integrated disease management, showed promising results in suppressing the pathogen and reducing infection levels within six to eight months [462]. Chemodiagnostic methods, including EDTA and TTC tests, along with physiological parameters like electrical conductivity and relative water content, can detect basal stem rot disease in coconut palms well in advance, up to 4 to 14 months before visible symptoms appear [463].

## 10. Disease Management Strategies to Control Basal Stem Rot in Other Crops

### 10.1. Cultural Practices

*Ganoderma lucidum* affects forests, particularly *Delonix regia* trees. However, *Peltophorum pterocarpum* shows tolerance to *G. lucidum*, even in semi-arid regions. Planting both species in alternating rows is recommended for the effective management and control of *G. lucidum* in plantations [464].

### 10.2. Biological Control

*Trichoderma koningi* and *T. viride* inhibited *Ganoderma* foot rot in Arecanut, with *T. viride*’s volatile metabolites showing the strongest inhibition. *Bacillus amyloliquefaciens* and *B. subtilis* also inhibited the fungi effectively. Volatile metabolites from both fungi and bacteria showed promise in controlling the disease [465]. Plant extracts, including garlic and black nightshade, showed a strong inhibition of *G. lucidum*, the cause of basal stem rot disease in arecanut. Other plant extracts, such as *Clerodendron infortunatum* and *Bidens pilosa*, also exhibited significant inhibitory effects. These findings indicate the potential of these plant extracts for controlling the disease [466]. *Trichoderma* spp. were tested as biological control agents against *Ganoderma* infection in Sengon seedlings. In vitro experiments showed that *Trichoderma* inhibited *Ganoderma* growth by 11.7% to 48.8%. Field trials revealed that seedlings treated with *Trichoderma* and organic materials had the highest height. *Trichoderma* and organic materials have potential for enhancing Sengon seedling growth and protection against *Ganoderma* attacks [467].

Three hundred and twenty-seven *Bacillus* spp. isolates from coconut plants were tested for antifungal activity against *G. applanatum* and *Thielaviopsis paradoxa*. Over 90% inhibited *G. applanatum*, and about 86% inhibited *T. paradoxa*. Thirteen *Bacillus* spp. were identified as potential biocontrol agents against these coconut pathogens [468]. Endophytic bacteria from coconut roots were isolated and evaluated for their potential in promoting rice seedling growth and inhibiting *G. lucidum*, the pathogen causing BSR in coconuts. Some strains showed positive effects on rice seedlings and the effective inhibition of *G. lucidum* growth. The study suggests their potential use for managing basal stem rot disease [469]. Basal stem rot disease threatens coconut cultivation in India. A study evaluated 29 plant products for managing the disease. Extracts of *Pungamia glabra*, *Azadirachta indica*, and *Prospis juliflora* inhibited *Ganoderma* growth in vitro. In the field, these extracts reduced the disease index and increased soil microorganism populations. These findings suggest the potential of these plant extracts for BSR management in coconut cultivation [470]. *Trichoderma viridae* was found in the tissue of *G. boninense*, the causal agent of root and bole rot in coconut. The coiling of *T. viridae* around *G. boninense* hyphae was observed, indicating its potential as a biocontrol agent against the pathogen [471].

### 10.3. Molecular Methods

Diagnostic methods for *Ganoderma* disease in coconuts were developed using ELISA and PCR techniques. Polyclonal antibodies and specific primers were used to detect the pathogen, and the spawn inoculum method provided early detection. Combining ELISA and PCR proved reliable for detecting *Ganoderma* infection in coconuts [472].

### 10.4. Integrated Disease Management (IDM)

Biocontrol agents, neem cake, hexaconazole root feeding, and micronutrient application effectively reduced basal stem rot disease in coconut palms caused by *Ganoderma* species. A talc-based formulation with *Trichoderma reesei*, *Pseudomonas fluorescens*, and neem cake significantly reduced the disease index. Mixed cropping with water yam suppressed *G. boninense* infection in oil palm seedlings, showing potential for disease management [417,473]. In the lower Assam districts of India, *Ganoderma* disease caused by *G. lucidum* is highly prevalent, with an incidence rate of up to 55%. The disease is spreading rapidly to other districts. A study suggests various management practices, including proper garden care, effective crop husbandry, phytosanitation measures, and the use of chemical and biological management strategies [474]. In India, *Trichoderma* spp. demonstrated effectiveness against *Ganoderma* disease in coconut gardens. The combination of *Trichoderma viride* and neem cake controlled *Ganoderma* spread, offering promising integrated management strategies [475]. Control methods including *Trichoderma harzianum*, neem cake, *Azotobacter*, and *phosphobacterium* were effective in managing basal stem rot disease in coconut palms. They reduced disease severity and increased nut yield, and tapping for neera production showed a lower disease index [476]. *Ganoderma* wilt disease, caused by *G. lucidum*, affects coconut palms in India. Soil temperature and moisture influence the severity of the disease. Organic manure, irrigation during summer, Bordeaux mixture, and root feeding with Aureofungin-sol and copper sulfate have shown effectiveness in controlling the disease [477].

## 11. Research on Butt Rot and Root Rot Diseases of Important Perennial Crops

### 11.1. Butt Rot

Butt rot, caused by the pathogenic fungus *G. zonatum*, is a destructive disease that affects a wide range of perennial crops, including oil palm, rubber, coconut, areca, and pygmy date palm [13]. It is characterized by the decay of the lower trunk or butt of affected plants, leading to significant structural damage and eventual death. *Ganoderma* species are known to infect the woody tissues of a tree, causing extensive decay and weakening of the tree’s support system [478]. The disease can have severe implications for plantation industries and natural forest ecosystems. It is known to spread through infected plant debris, spore dispersal, and root-to-root contact [279] Effective management strategies for *Ganoderma* butt rot involve a combination of cultural practices, chemical control, and the use of resistant planting materials [332]. The early detection and timely implementation of control measures are crucial for minimizing the impact of the disease and preserving the health and productivity of affected trees [13].

Further research is ongoing to understand the biology, epidemiology, and management of *Ganoderma* butt rot. Studies focus on aspects such as the life cycle of the fungus, transmission mechanisms, host–pathogen interactions, and the development of sustainable and integrated disease management approaches. Overall, the study and management of *Ganoderma* butt rot are essential for protecting tree health, maintaining ecosystem stability, and sustaining the productivity of important perennial plantations.

### 11.2. Red Root Disease

Red root rot disease is a major concern in Southeast Asia’s tropical plantation/estate forests, affecting their long-term sustainability [25]. *Ganoderma philippii* has been identified as the main pathogen responsible for this disease, which spreads through physical contact and spore dispersal. Although the disease progresses slowly, it eventually leads to plant death [479]. Integrated disease management strategies focus on plant resistance and biocontrol measures [207]. Efficient screening protocols have been developed to identify resistant plant materials, while effective biocontrol agents have been isolated [25]. In the upcoming section, we delve into the identification, dispersal, and control of *Ganoderma* root rot disease. This discussion will provide valuable insights into the characteristics of the disease and offer strategies for its management.

*Acacia mangium* and *Eucalyptus pellita* are susceptible to red root rot disease caused by *Ganoderma philippii* and *Pyrrhoderma noxium*, leading to the failure of *A. mangium* as a commercial pulpwood in Indonesia. Soil type influences root infection, with higher mortality in soils with a low clay content. Effective measures are needed to prevent root rot diseases in existing *E. pellita* plantations [480]. *Ganoderma philippii* is a destructive root rot pathogen in *Acacia mangium* plantations. The biology, spread, and management strategies of *G. philippii* were investigated, including basidiospore germination, breeding systems, and population structures. Disease severity increased with rotation, emphasizing the need to control pathogen spread for effective management [481]. *Ganoderma philippii* exhibits genetic diversity and clonal spread in *A. mangium* plantations. Measures to prevent underground pathogen spread and basidiospore infections are essential for managing root rot [479].

DNA sequence variation in the rDNA internal transcribed spacers (ITSs) was utilized to develop species-specific primers for *G. philippii* and *G. mastoporum*. These primers were validated against other fungal species. PCR tests using these primers successfully identified major root rot pathogens in *A. mangium* and *Eucalyptus pellita* plantations, providing a reliable tool for disease management [482]. A survey in a Nigerian rubber clonal garden found a 0.02% infection rate of red root rot. Two *G. psuedoferreum* varieties were observed: one infective on live trees and the other non-infective on dead trunks. This study underscores the need for routine inspections and control measures to curb pathogen spread in rubber plantations [483]. Damar trees (*Agathis* spp.) in Mount Walat forest Education (HPGW) face a significant threat from red root rot disease caused by *Ganoderma*. Urgent attention is needed to address this issue, and further research is necessary to understand disease factors and potential alternate hosts [467].

*Ganoderma* root rot is a significant disease in *Acacia mangium* plantations. *Ganoderma philippii* has been identified as the primary pathogen causing the disease, supported by phylogenetic and morphological analyses. Management strategies should focus on targeting *G. philippii* [484]. *Ganoderma* has caused significant damage to *Acacia mangium* plantations in Sumatera and Kalimantan, as well as rubber and oil palm plantations in Indonesia. The accurate identification of the pathogen is crucial for disease control, and this study successfully identified *G. philippii* using morphological characteristics and confirmed it with DNA sequencing. Other fungi, including *G. australe* and *Phellinus* spp., were also found to attack *Acacia mangium* roots [485]. Red root rot caused by *G. philippii* affects *Acacia mangium*, leading to crown dieback and tree death. Disease spread and progression are related to initial root rot incidence. Surveys should consider above-ground symptoms and root infection extent. Remote sensing may be explored for future disease surveying [486].

### 11.3. Other Root Diseases

*Ganoderma* root rot disease was assessed at Universitas Gadjah Mada (UGM) campus, Indonesia, showing an increase in affected trees and disease incidence. The occurrence of the disease remained rare (<10%), with crown damage being more prominent in the closed system and among legume trees. The disease exhibited a clustered spread pattern, with some random and uniform patterns observed in specific plant groups and locations [487]. *Ganoderma lucidum* is a particularly significant pathogen associated with the large-scale death of Shisham trees by root rot (*Dalbergia sissoo*) [488]. UGM’s arboretum in Yogyakarta has experienced tree decline and death from *Ganoderma* sp. attacks since 2015. A survey from 2018 to 2021 found an increase in *Ganoderma* infection, affecting 1.57% of trees in 2021. *Pterygota alata*, *Pterocarpus indicus*, and *Adenanthera pavonina* were the most affected species [489].

Various *Ganoderma* isolates, including *G. lucidum*, *G. boninense*, and *G. applanatum*, displayed virulence on *Acacia mangium* seedlings, leading to infection symptoms and reduced seedling weight. *G. lucidum* showed the highest virulence. These results underscore the risk posed by these *Ganoderma* species as root rot pathogens for *A. mangium*, with palm oil and infected rubber plants serving as possible sources of infection [68]. *Ganoderma* root and butt rot disease has led to the death of *Jacaranda mimosifolia* trees in Pretoria, South Africa. Two new species, *Ganoderma enigmaticum* and *Ganoderma destructans*, were also discovered as the primary causes of the disease [490].

A root rot disease outbreak in an organic Assam tea plantation was caused by *G. australe*, and it exhibited pathogenic characteristics and caused necrosis in tea seedlings [491]. A new disease affecting mango trees in Ismailia Governorate, Egypt, has been linked to *Ganoderma* sp. The disease causes basal rot, bracket formation, and tree death. Pathogenicity tests confirmed the ability of *Ganoderma* sp. to cause root rot and dieback in mango trees [492]. *Ganoderma* species associated with root rot disease in *Acacia mangium* and *Eucalyptus pellita* plantations in Indonesia were identified through morphological and DNA analyses. *G. philippii* was the most prevalent species, along with *G. mastoporum*, *G. australe*, *G. subresinosum*, and *Ganoderma* sp. The cultural characteristics of these isolates were described [493]. *Ganoderma lucidum* causes root rot and mortality in *Dalbergia sissoo* trees. A clonal seed orchard and seed production area were established in northern India, and, after 10 years, five highly resistant and five susceptible clones were identified based on growth form, disease resistance, and disease development [494].

*Ganoderma lucidum* causes root and butt rot disease in *Cassia nodosa*, *Cassia fistula*, and *Delonix regia* trees. The disease is lethal, particularly affecting young trees, which can die within 6 to 24 months of symptom onset. *Cassia fistula* is the most susceptible, with trees dying within six months of the first visible signs. *Ganoderma lucidum* shows a preference for these host trees, infecting them while sparing other tree species nearby [495]. *Ganoderma lucidum* was identified as the cause of root rot disease in jojoba plants under arid conditions in India. Infected plants exhibited symptoms such as drying twigs, yellowish-brown leaves, and complete drying up within a few months. Basidiocarps developed near decaying roots, confirming the pathogenicity of the fungus. Inoculation experiments with healthy jojoba plants further confirmed the role of *Ganoderma* in causing root rot [496]. *Ganoderma lucidum* caused root rot in an *Acacia arboretum*, primarily infecting stumps. *Acacia albida*, *A. aneura*, *A. decurrens*, *A. murrayana*, and *A. victoriae* were highly susceptible to the disease, while *A. greggii* and *A. verek* showed resistance [497]. The root rot of neem (*Azadirachta indica*) caused by *G. applanatum* is a serious disease and is recorded here for the first time from the Garo Hills of Meghalaya [498]. A high mortality rate due to *G. lucidum* was reported in young khair plantations (*Acacia catechu* Willd.) established as reforested stands. Up to 55 percent of plants were killed in 9-year-old plantations [477]. A newly identified grapevine syndrome called “colapso” is characterized by rapid wilting, yellowing, and death within three days. It is associated with a specific *Ganoderma* fungus species found on decaying wooden posts near affected vines. Overhead trellising systems promote *Ganoderma* growth. This is the first documented association between *Ganoderma* and grapevine disease [499].

## 12. Control Measures of Butt Rot, Root Rot, and Other Root Diseases of Important Perennial Crops

### 12.1. Biocontrol

The PSB isolate EF.NAP 8 demonstrated biocontrol potential against *G. philippii* and *Fusarium oxysporum* 0148c, pathogens causing root rot in acacia plants. It inhibited the growth of *G. philippii* by 34.44% and that of *F. oxysporum* 0148c by 33.33%. These findings suggest EF.NAP 8 as a potential agent for controlling these fungal pathogens in acacia plants [500]. Ginger and lemongrass showed a high disease incidence, while arrowroot, turmeric, and temulawak demonstrated a low disease incidence in suppressing *Ganoderma* root rot in oil palm nurseries. The growth of oil palm seedlings was influenced by soil properties and *Ganoderma* presence. Leaf color changes were observed due to *Ganoderma* infection. These findings highlight the potential of antagonist plants in managing *Ganoderma* sp. root rot disease [501].

The native biocontrol agents *Trichoderma* spp. and *Aspergillus* spp. show promise in controlling *G. lucidum* root rot in Indian mesquite trees. They inhibit *G. lucidum* growth and can be combined with insecticides for effective management [305]. White-rot fungi were screened for their potential as biological control agents against *Ganoderma* root rot disease in *Acacia* plantations. Out of 107 samples, 2 isolates demonstrated the ability to overgrow *G. philippii*, offering promising options for control [502].

*Bacillus* spp. isolates from coconut rhizospheric soil and roots in India showed strong antifungal activity against *G. applanatum* and *Thielaviopsis paradoxa*. Multiple mechanisms, including chitinase production and antibiotics, were identified. Thirteen *Bacillus* spp. have potential as biocontrol agents against these coconut pathogens [468]. *Ganoderma lucidum*, the coconut wilt pathogen, was isolated from coconut trees’ basal stems and rhizosphere soil. The study explored the antifungal activity of *Eichhorinia crassipes* plant extracts against *G. lucidum.* Two concentrations (150 mg/mL and 300 mg/mL) were tested, with the higher concentration showing the most effective inhibitory activity [494]. Fungi antagonists, including *Trichoderma* and *Aspergillus* species, showed a significant inhibition of *G. pseudoferreum* in rubber plantations. These fungi have potential as biocontrol agents against red root disease [503].

### 12.2. Molecular Methods

The genetic diversity of *Ganoderma* spp. causing root rot in cacao and shade trees was assessed using a RAPD analysis. Out of 45 samples, 220 polymorphic DNA fragments were obtained. A cluster analysis revealed high variability among the samples, with three distinct groups being identified. This information is important for developing control strategies against *Ganoderma* disease [504].

### 12.3. Integrated Disease Management (IDM)

An integrated management approach combining chemical fungicides, biocontrol agents (specifically *Trichoderma harzianum*), and selected botanicals showed promise in suppressing *G. lucidum*, the causative agent of root rot. Bavistin, propiconazole, mancozeb, and certain botanicals demonstrated effectiveness against the fungus [505]. A screening method was developed to assess red root rot tolerance in *Acacia* species affected by *G. philippii*. It allows for faster experiments and reveals variations in tolerance and susceptibility. The objective is to develop resistant *Acacia* genotypes and understand tolerance mechanisms [506]. A faster and more reliable shade house screening method for red root rot tolerance in *Acacia mangium* has been developed, allowing for the selection of tolerant materials. Field experiments and commercial plantations have demonstrated that these tolerant materials experience a lower root rot incidence. This presents an opportunity for the integrated management of red root rot in tropical acacia plantations by incorporating plant tolerance [507]. 

*Ganoderma* root rot is a widespread problem causing host death. Accurate identification is challenging due to morphology-based issues. A molecular analysis aids in taxonomy and identifying specific pathogens. The *Ganoderma lucidum* complex caused disease in *Jacaranda mimosifolia* [508]. Indian mesquite trees in arid regions of India are dying due to an insect–disease association involving *Acanthophorus serraticomis* insects and *G. lucidum* pathogens. A practical management strategy using biocontrol agents and agricultural waste residues shows promise in controlling *Ganoderma* and promoting tree growth [509]. A *Ganoderma* species causing root disease in lowland tea was identified, and fumigation with methyl bromide proved effective in controlling the disease, preventing *Ganoderma* colonization in tea stumps by enhancing soil microflora. Mulching with NPK fertilizer further improved disease management. The problem in lowland tea areas and the efficacy of methyl bromide fumigation are emphasized [510].

## 13. Future Challenges

In the context of future challenges in addressing the menace of phytopathogenic *Ganoderma* species, several critical aspects demand meticulous attention and scientific endeavor.

Foremost among these challenges is the perpetual threat of emerging pathogenic strains. As *Ganoderma* evolves, understanding these species’ genetic and phenotypic variations is imperative. This necessitates comprehensive surveillance and genetic studies to detect new strains promptly and assess their pathogenic potential. It also underscores the urgency of developing effective strategies to combat these evolving pathogens. Climate change, characterized by shifting weather patterns and temperatures, profoundly influences the prevalence and distribution of *Ganoderma* infections. A challenge must be met to unravel the intricate relationship between these changing environmental factors and disease dynamics. This understanding is vital for the adaptation of control measures and the development of climate-resilient strategies. Sustainability, a cornerstone of contemporary agriculture and forestry, remains a focal point. While integrated approaches have proven effective, the quest for ecologically sound and sustainable control measures persists. This entails continuously exploring innovative techniques, such as biological controls and environmentally friendly materials, to manage *Ganoderma* sustainably. The pursuit of host resistance presents an ongoing scientific challenge.

Identifying and breeding plant varieties with enhanced resistance against *Ganoderma* species is pivotal. This endeavor requires rigorous screening for resistance traits, the development of molecular markers, and the broad dissemination of resistant cultivars across various crops. It is a research frontier that necessitates unwavering commitment. International collaboration stands as a central pillar in our pursuit of overcoming *Ganoderma*’s threats. Given the global nature of the problem, collective scientific efforts on an international scale are paramount. Collaboration among scientists, breeders, and growers from different regions is indispensable. This collaboration encompasses sharing knowledge, harmonizing research efforts, and establishing international networks dedicated to addressing the multifaceted challenges posed by *Ganoderma*. Hence, as we confront these multifaceted challenges associated with phytopathogenic *Ganoderma*, a rigorous scientific approach remains our guiding beacon. Rigorous genetic studies, climatic research, sustainable agricultural practices, and international collaboration are essential components of the strategy to combat this persistent threat. With unwavering commitment and shared scientific endeavor, we are poised to meet these challenges head-on and protect global agriculture and forestry from the pernicious influence of *Ganoderma*.

## 14. Conclusions

In conclusion, our comprehensive analysis of phytopathogenic *Ganoderma* has shed light on the significant challenges that these pathogens pose to global plant health, with far-reaching economic consequences in agriculture and forestry. Through this review, we have delved into the intricacies of their life cycle, their disease mechanisms, and the environmental factors that underpin their spread. We have explored the nuanced interactions between *Ganoderma* and their host plants, deciphering the pathogen’s strategies for successful colonization and the subsequent activation of plant defense responses. One of the key takeaways from our analysis is the importance of adopting an integrated approach to managing *Ganoderma*-related diseases. By amalgamating cultural practices, harnessing biological control agents, judiciously employing chemical treatments, and cultivating resistant plant varieties, we can significantly enhance the effectiveness and sustainability of our control strategies. This review underscores the necessity of a collaborative effort involving scientists, breeders, and growers to collectively devise and implement strategies that can stand up to the challenges posed by this devastating plant pathogen. The insights provided here serve as a foundation for future research, paving the way for innovative and sustainable approaches to combat phytopathogenic *Ganoderma*. As we continue to learn, adapt, and work together, we promise to reduce these pathogens’ impact on our agricultural and forestry systems, ultimately securing a more resilient and productive future for our plants and crops.

## Figures and Tables

**Figure 1 jof-10-00414-f001:**
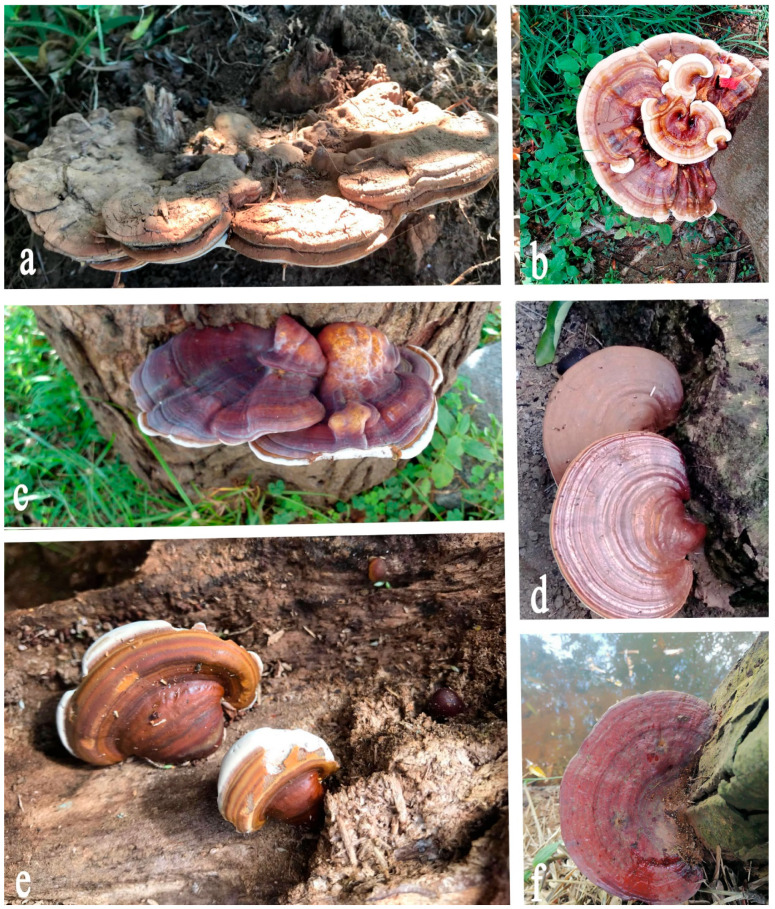
Pathogenic and wood-decaying *Ganoderma* species (https://www.inaturalist.org/, acceced 1 April 2024). (**a**) *G. applanatum*. (**b**) *G. curtisii.* (**c**) *G. resinaceum*. (**d**) *Ganoderma* sp. (**e**) *G. tsugae*. (**f**) *Ganoderma* sp.

**Figure 2 jof-10-00414-f002:**
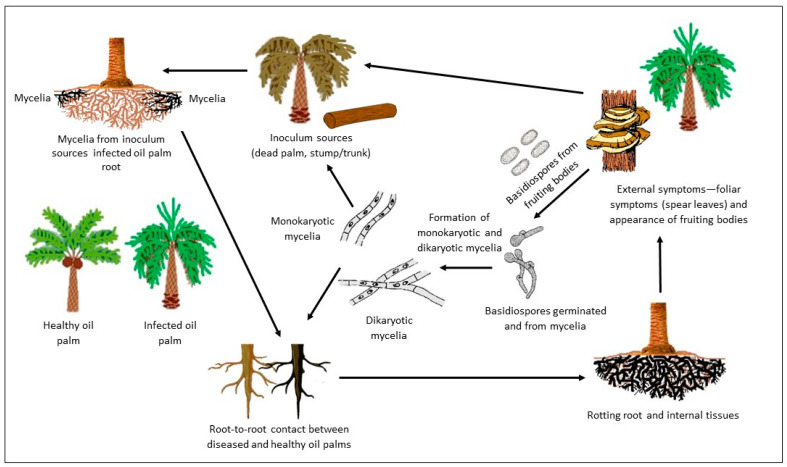
*Ganoderma boninense* life cycle in oil palm.

**Figure 3 jof-10-00414-f003:**
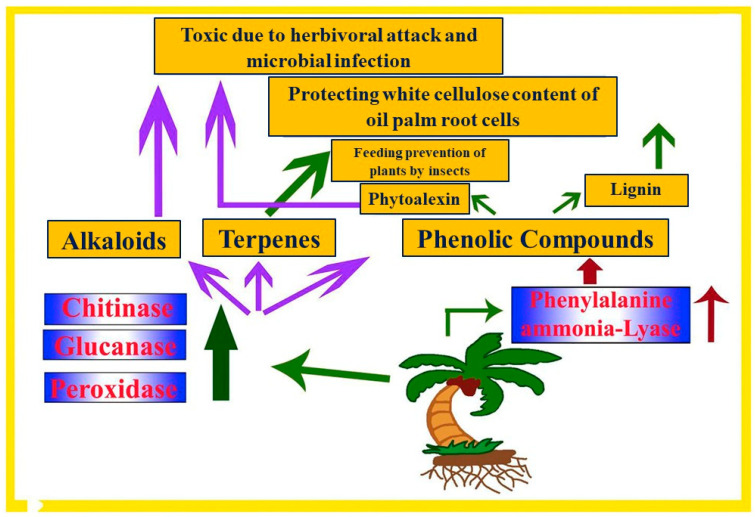
Role of secondary metabolites in plant defense against *G. boninense* in oil palm.

**Table 1 jof-10-00414-t001:** Molecular methods for detection of *Ganoderma*, along with their corresponding primer sequences.

Method/Technique	Responsible Primer Sequence		References
Real-Time PCR	ANTHO (5′-ACAACATGGTCCCCGGTCT-3′, 3′-GGTGGAGGATGCTCTTGTAGGT-5′), LEUCO (5′-ACAACATGGTCCCCGGTCT-3′, 3′-CGGCGGACTTTCCTCTTTTC-5′), ETHYLENE (5′-AAGAGCAAGGCAGGGAATGG-3′, 3′-CTTCTGCGCTGTCAAAGGTTC-5′), MANNOSE (5′-TCGGATGGGAACCTTGTGG-3′, 3′-CCGATCTCGTTGGAGGATACAG-5′), CHALCONE (5′-GAGCAGATCCAATGCAAGGTGT-3′, 3′-GGTTGAGGAGGTGGAAGGTGA-5′), SENESCENCE (5′-GGCACGGCCATCAGTAGAGTA-3′, 3′-AGCCAAGCGTTCATAGCGAC-5′), THAUMATIN (5′-ACGAGGGAGATGTCGATGAA-3′, 3′-GACTGCGGTGGTAAACTTGC-5′)		[20]
qPCR-Gene Characterization (*ERG11*)	α-Tubulin (GTR7: 5′-GCACCGACTCTG GTGATG CT-3′, GTR8: 3′-GATAGG CTATGGTCG CGA AG-5′, β-Tubulin (GBR3: 5′-GAGTTCACTGAG GCC GAGTC-3′, GBR4: 3′-TGCAACACG CTTATTCTTCG-5′, eEF2 (GER1: 5′-TGGTCA AGA ACATCC GTAT-3′, GER2: 3′-CGC TAACAA AGACAAGGG -5′, ERG11 (GERR3: 5′-TCGTAG GAAGTTCGAGTGAGTA-3′, GERR4: 3′-ATC CAT TCATTACGTAGTGTGC-5′		[193]
Real-Time PCR Assays	GB4: 5′-CGTTCGTTTGACGAG TTTGC-3′, 3′-GGTTGGTTTCTTTTCCT-5′		[164]
PCR with SSR Markers	KT124397: 5′-CGCCATGCCCACCACCAGAG-3′, 3′-GACCCGGCTGCCCGAATGAG-5′ KT124402: 5′-ACAAGGCTCAAGGCAGCGCA-3, 3′-GCACACCCCAGCAACAGGAGG-5′ KT124403: 5′-GGCGACGAGGGCACGAGAGA-3′, 3′-CCGCACTTTCGCCAACCACC-5′ KT124400: 5′-AGCTCCCCTCCCAGCTCCAAC-3′, 3′-GAATGCGGCGGGGAAACGGA-5′ KT 124399: 5′-GCACAGGCACAAGCGCAAGG3′, 3′-CGACGACCGCCCCAAAGGAT-5′ KT 124394: 5′-CGGGAAGTGGTGAACGGTGGT-3′, 3′-GGGTGGCTTGACAGCGGCAT-5′		[165]
LAMP (Loop-Mediated Isothermal Amplification)	F3: 5′-GCTGGCGTGATCTCCTCTA-3′ B3: 5′-AACGTGCCTGGTATGGAATT-3′		[168]
PCR-Based Diagnostic Method	Gan1: 5′-TTG ACT GGG TTG TAG CTG-3′, Gan2: 5′-GCG TTA CAT CGC AAT ACA-3′, Gan ET: 5′-GAG TTG TCC CAA TAA C-3′ ITS3: 5′-GCA TCG ATG AAG AAC GCA GC-3′		[174,185,194]
PCR and RAPD	PCR	Gan1: 5′-TTG ACT GGG TTG TAG CTG-3′ Gan2: 5′-GCG TTA CAT CGC AAT ACA-3′ ITS 1: 5′-TCCGTAGGTGAACCTGCGG-3′ ITS 4: 5′-TCCTCCGCTTATTGATATGC-3′	[176]
	RAPD	OPI 01: ACCTGGACAC, OPI 07: CAGCGACAAG OPI 12: AGAGGGCACA, OPI 14: TGACGGCGGT	
Multiplex PCR	-		[177]
DNA Sequence Analysis	ITS1: 5′-TCC GTA GGT GAA CCT GCG G-3′ and ITS4: 5′-TCC GCT TAT TGA TAT GC-3′		[76,179]
qPCR	ITS1 (5′-TCC GTA GGT GAA CCT GCGG-3′) and ITS4 (5′-TCC TCC GCT TAT TGATAT GC-3′)		[180]
RAPD and PCR-RFLP Techniques	RAPD	CRL-1 (5′-CCAGCGCCCC-3′), CRL-2 (5′-CTGCCGCCGC-3′), CRL-7 (5′-GCCCGCCGCC-3′), CRL-11 (5′-CCACCGCGCC-3′) and CRL-34 (5′-GACCGCGCCC-3′)	[183,187,191,192]
	PCR-RFLP	ITSI (5′-TCC GTA GGT GAACCTGCG G-3′) and ITS4 (5′-TCC TCC GCT TAT TGA TAT GC-3′)	

**Table 2 jof-10-00414-t002:** Summary of control methods for basal stem rot in oil palm.

Category			Practice		Findings	References
Agricultural Practices	Co-Infection		Co-infection with different strains of *G. boninense* in oil palm seedlings		Showed similar symptoms with less severity than a single aggressive strain, along with higher fungal colonization in oil palm roots.	[263]
	Field Trials		Different practices to reduce *Ganoderma* infection in replanted oil palms		Fallowing, poisoning, and windrow 2:1 system reduced infection rates to 3%. Fallowing alone in the same system resulted in 6% infection rates.	[264]
Cultural Practices	Land Management		Fallowing and proper plant placement		Fallowing reduced BSR disease in oil palm by avoiding infected remnants and exposing soil to sunlight.	[265]
	Crop Rotation		Rotation with sugarcane		Studied as a cultural practice for managing BSR disease in oil palm production. Shows potential for disease management while maintaining productivity.	[266]
	Mixed Planting		Planting with edible herbaceous perennial plants		Oil palm seedling growth suppressed; minimal impact on *Ganoderma* infection; Arrowroot showed strongest competition, followed by canna and cocoyam, while water yam had minimal interference.	[267]
	Trenching		Creating trenches around infected palms on peat		Effectively minimized spread to neighboring palms. Mounding-infected palm’s base extended its lifespan.	[268]
	Sanitation		Controlling old palm remnants		Crucial in BSR management. Palm size, growth vigor, and inoculum size impact disease. Early control and preventing root contact with infected trunks vital.	[270]
	Poisoning		Poisoning old palms before felling them		Significantly reduced *Ganoderma* disease in oil palm replanting. No significant difference in disease levels between underplanting and complete felling methods.	[271]
Biological Control	*Trichoderma* spp.	Strains TSU and TGLP	Act as potential biocontrol agents		Strong inhibition of *G. boninense* and produced chitinase, glucanase, and indole acetic acid.	[272,273]
					High antagonistic activity against *Ganoderma*.	[283]
		*Trichoderma* spp. mixture	As a sustainable solution for managing basal stem rot in oil palm		Reduced disease symptoms, enhanced plant defense mechanisms, and slowed down disease progression.	[285]
		TR1 biocontrol product	As an eco-friendly alternative to fungicides for *G. boninense* control		Achieved effective control of *G. boninense* in oil palm by reducing colonization, pathogen re-isolation, and ergosterol content in field experiments.	[108]
		T9 and T29	Potential biocontrol for managing BSR in oil palm seedlings		Achieved effective suppression of BSR disease.	[292]
		*Trichoderma* spp. gene modulation	Acts as a biofertilizer, improving nutrient status		Modulated genes related to hormone production, antioxidant systems, and cell wall metabolism.	[284]
	*Trichoderma asperellum*		Potential as a biocontrol agent against *Ganoderma* spp.		Effectively inhibited mycelial growth of *Ganoderma* species.	[274]
		Endophytic strain M103	Strong potential as a biocontrol agent against *G. boninense*		Inhibited mycelial growth of *G. boninense*.	[275]
		Endophytic strains ET537 and ET523			Suppressed *G. boninense* infection and increased total phenolic content, enhancing resistance to BSR disease.	[278]
		*T. asperellum* on different media compositions	Strong potential as biocontrol agents against *G. boninense*		Achieved strong inhibition of *G. boninense*.	[277]
			Antagonistic properties against *G. boninense*		Inhibited *G. boninense* growth and exhibited traits like phosphate solubilization, indole acetic acid production, siderophore production, and hydrolytic enzyme production.	[117]
		Metal-tolerant Endophyte strain LF11	Controlling GB in metal-laden soils		Showed inhibition of *G. boninense* in vitro.	[280]
		Strain UPM1/*Bacillus*	Potential for sustainable BSR management in oil palm		Combination treatment effectively mitigated BSR disease in oil palm.	[316]
	*T. harzianum*	Endophytic strain M108	Strong potential as a biocontrol agent against *G. boninense*		Inhibited mycelial growth of *G. boninense*.	[275]
		Strain AC2	Acts as a soil microbial antagonist			[276]
		Pre-inoculation of oil palm seedlings			Suppressed *Ganoderma* disease, promoted seedling growth, and reduced disease severity.	[272]
		Strain FA 1132	Strong potential as a biocontrol agent against *G. boninense*		Showed highest effectiveness in reducing disease severity	[287]
			*Trichoderma*-incorporated surface mulch offers an eco-friendly alternative to chemical pesticides		Achieved symptom-free oil palm plants with lower disease severity and increased root and leaf weights.	[92]
		In vitro growth study	Potential as a biocontrol agent against *Ganoderma* spp.		Achieved highest inhibition of *Ganoderma* sp. isolates.	[290]
			Holds promise as a biocontrol agent against *G. boninense* in oil palm		Suppressed *G. boninense* growth both in vitro and in vivo.	[291]
		PSA culture media	Holds promise as biocontrol agents against *G. boninense*		Enhanced spore production and antagonistic activity against *Ganoderma.*	[293]
			Potential role in enhancing defense mechanisms against microbial pathogens		Enhanced chitinase activity in oil palm tissues infected with *G. boninense*.	[294]
	*T. reesei*	Endophytic strain ET501	Suppresses *G. boninense* infection		Increased the total phenolic content and enhanced resistance to BSR disease.	[278,281]
			Exhibits high antagonistic and antibiosis activity		Promising candidate for oil palm disease management.	
	*T. virens*		Efficacy study in oil palm pre-nurseries		Reduced disease intensity caused by *G. boninense* and promoted stem diameter growth in oil palm seedlings grown on peat medium.	[279]
			Promise as biocontrol agents against *G. boninense*		Ethyl acetate extract and specific fractions displayed strong antifungal activity against *G. boninense.*	[289]
		PSA culture media	Promise as biocontrol agents against *G. boninense*		Enhanced spore production and antagonistic activity against *Ganoderma.*	[293]
	Other Microbes and Macrofungi	*Burkholderia*	*Burkholderia gladioli* Strain	Acts as a soil microbial antagonist		[276]
			*Burkholderia* spp. strains	Promise as biocontrol agents against *G. boninense*	Exhibited antifungal properties and hydrolytic enzyme activities against *G. boninense.*	[330]
			GanoEB2 in bioorganic powders	Potential biocontrol agent	Suppressed *G. boninense* effectively in oil palm seedlings.	[347]
		AMF		Potential as biological control agents against *G. boninense* in oil palm cultivation	Show potential as biocontrol agents against *G. boninense* in oil palm cultivation.	[272]
				Potential in enhancing oil palm seedlings’ resilience to *Ganoderma* sp. infection	Promoted growth and resistance of oil palm seedlings against *Ganoderma* sp. infection.	[321]
				Suppress *Ganoderma* diseases	Promoted bole diameter growth in oil palm nurseries.	[322]
				Potential as biological control agents against *G. boninense*	Improved plant growth and chlorophyll content in *Ganoderma*-infected seedlings.	[334]
			AMF and endophytic bacteria	Improve biocontrol efficacy	Significantly reduced BSR disease development in nursery and field trials.	[345]
		*Bacillus*	Strain UPM15	Pre-inoculation of oil palm seedlings	Suppressed *Ganoderma* disease, promoted seedling growth, and reduced disease severity.	[19]
			*Bacillus subtilis*	Acting against *G. boninense*	Enhanced growth, increased salicylic acid, and produced indole-3-acetic acid.	[333]
			*Bacillus cereus*	Potential as a natural control method	Strong antifungal activity against *G. boninense.*	[336]
		*Pseudomonas aeruginosa*	Antagonistic properties against *G. boninense*		Inhibited *G. boninense* growth and exhibited traits like phosphate solubilization, indole acetic acid production, siderophore production, and hydrolytic enzyme production.	[117]
			Encapsulated *P. aeruginosa* makes a promising strategy for managing *Ganoderma* diseases in the oil palm industry		Showed strong bioactivity against *G. boninense*; encapsulation improved biocontrol efficacy.	[313]
		*Diaporthe miriciae* strain LF9	Controlling *G. boninense* in metal-laden soils		Showed inhibition of *G. boninense* in vitro, improved tolerance, and reduced disease severity in oil palm seedlings.	[280]
		*Streptomyces*	*Streptomyces nigrogriseolus* GanoSA1	Induction for biological control against *G. boninense*	Induced production of pathogenesis-related proteins, including β-glucanase, peroxidase, polyphenol oxidase, and phenylalanine lyase in oil palm seedlings.	[310]
				Potential as a biocontrol agent	Showed strong inhibition of *G. boninense.*	[329]
			*Streptomyces gelaticus* isolates AKT19, AKT28, and AKT52	Potential as biological control agents against *G. boninense*	Showed strong suppression of *G. boninense* growth in vitro; produced volatile organic compounds and bioactive compounds inhibiting *G. boninense* growth.	[318]
			*Streptomyces palmae* CMU-AB204T		Reduced disease severity, enhanced plant vigor, and had antimicrobial compounds inhibiting *G. boninense* and other pathogens.	[309,328]
			*Streptomyces* spp. strain A19		Showed strong inhibition of *G. boninense* growth; metabolites caused damage to its hyphae.	[204]
			*Streptomyces*-like actinomycetes	Potential as biocontrol agents	Showed significant inhibition of *G. boninense.*	[338]
		Keratinolytic fungi A 12 and A 18	Potential as biocontrol agents for agriculture		Showed strong antagonistic ability against *G. boninense*, inhibiting its growth and causing abnormalities in hyphae.	[314]
		Ligninolytic fungi from arrowroot, cocoyam, and canna plants	Potential for managing basal stem rot disease in oil palm plantations		Inhibited *G. boninense* wood decay and reduced root infection in oil palm.	[315]
		Actinobacteria and Firmicutes from oil palm rhizosphere	Potential as biological control agents against *G. boninense*		More abundant in asymptomatic trees’ rhizosphere soil	[317]
		*Hendersonia toruloidea* (GanoEF1)			Suppressed BSR disease caused by *G. boninense* in oil palm seedlings.	[319]
		Indigenous bacteria from peatlands	Potential as biocontrol agents		Exhibited inhibitory effects on *Ganoderma* growth.	[458]
		Algal compounds	Potential in inhibiting *Ganoderma* metabolism		In silico analysis identified serotonin, 5-methoxytryptamine, and melatonin as compounds with strong binding affinity to target.	[323]
		Bacterial isolates from peatlands	Potential as biocontrol agents for BSR disease in oil palm cultivation on peatlands		Showed antagonistic activity against *Ganoderma* under low pH conditions.	[324]
		Mycolytic enzyme-producing bacteria	Potential to be effective biocontrol agents for managing basal stem rot disease in oil palm		Exhibited antifungal properties against *G. boninense.*	[325]
		Soil microbiome			Lower BSR incidence associated with higher microbial diversities and disease-suppressive bacteria.	[326]
		*Talaromyces apiculatus* (Cr) and *Clonostachys rosea* (Ta)	Co-inoculation		Enhanced leaf area, biomass, nutrient content, and disease control.	[327]
		*Scytalidium parasiticum*	Potential as biocontrol agents		Produced antimicrobial compounds that help combat *Ganoderma* infections.	[331]
					Reduced *Ganoderma* infection and disease severity while promoting plant growth.	[343,344]
		Fungi from peatlands	Potential control agents for BSR		Showed antagonistic activity against *Ganoderma* at acidic pH.	[332]
		Seaweed extracts (*Caulerpa racemosa* var. *lamouroxii*)	Potential as biocontrol agents for BSR disease		Showed strong antifungal activity against *G. boninense* with identified bioactive compounds.	[459]
		Endophytic bacteria			Inhibited *Ganoderma* growth and reduced disease development in nursery trials.	[342]
		Actinomycetes isolates	Potential as biocontrol agents		Showed significant inhibition of *G. boninense.*	[346]
		*Cladobotryum semicirculare*			Inhibited *G. boninense* growth.	[358]
		White-rot hymenomycetes	Show antagonistic potential against *G. boninense*.		Promoted better growth in oil palm seedlings.	[340]
		*Lentinus Cladopus* LC4			Potential biological control agent against *G. boninense* despite no inhibition zone.	[341]
	Other Methods	Phenolic compounds	As a sustainable method, reduce reliance on synthetic fungicides and minimize environmental impact		Suppressed the pathogens and their impact on the cell membrane potential.	[362]
			Inhibitory effects on *G. boninense* enzymes		Benzoic acid demonstrated highest effectiveness and shows promise as an alternative to chemical inhibitors for controlling *Ganoderma* infections.	[368]
		Hendersonia GanoEF Biofertilizer	Potential as biocontrol agents		Controlled *Ganoderma* disease in oil palm seedlings, resulting in improved seedling growth and reduced disease incidence and severity.	[363]
		GanoCare^®^ Fertilizer Technology	Enhances the seedlings’ defense system against BSR.		Showed efficacy in suppressing BSR, improved growth parameters, and reduced BSR incidence and severity in oil palm seedlings.	[364]
		Crude phenazine synthesized by *P. aeruginosa* UPMP3	Shows promise as a biocontrol agent against BSR in oil palm		Reduced disease severity and improved plant vigor; upregulation of specific genes suggests the involvement of induced resistance in controlling the pathogenic *G. boninense.*	[365]
		Colonized System of *Ganoderma* Vaccine (CHIPS^®^)	As a biocontrol agent against *Ganoderma* disease in oil palm		Showed a significant decrease in disease severity index (DSI) compared to other treatments, offering a higher profit ratio.	[366]
		Biological control agents (BCAs)	BCAs hold promise for *Ganoderma* control in oil palm		BCAs did not prevent BSR infection in nursery trials but disrupted the pathogen’s progression based on ergosterol content.	[367]
					Three BCA products successfully reduced *G. boninense* colonization in oil palm trials, lowering ergosterol content and disease incidence (DI), decreasing ergosterol levels in trunk tissues.	[373]
		CuSO_4_ treatment	Enhancement of antifungal activity		Increased phytochemical content and enhanced antifungal activity against *G. boninense.*	[369]
		Organic fungicide	As a biocontrol agent against *Ganoderma* disease in oil palm		Promoted palm root growth, improved plant performance, induced flower formation, and increased nutrient levels. Fruit production and oil content were higher with organic fungicide treatments.	[371]
		Papaya leaf extracts	Potential to hinder the growth of *G. boninense*		Exhibited antifungal properties against *G. boninense*. Methanol and acetone extracts were effective in inhibiting fungal growth.	[372]
		ABM-OP1 (Advanced Beneficial Microbialsystem) enzyme-based product	A promise for controlling *Ganoderma* infection and prolonging oil palm tree productivity		Effectively destroyed over 85% of internal *Ganoderma* mycelia in oil palm.	[374]
Chemical Control	Nanofungicides	Dazomet–micelle nanodelivery systems (DAMINs) as a surfactant have the potential to be effective nanofungicides for disease control			Showed the highest inhibitory activity against *G. boninense.*	[375]
	Nanodelivery Systems	Anionic surfactant-based nanodelivery systems showing potential as effective agronanofungicides for managing basal stem rot disease in oil palm			Demonstrated stronger antifungal effects against *G. boninense.*	[376]
		Fungicide nanodelivery agents based on hexaconazole–micelle systems show potential for combating basal stem rot disease caused by *G. boninense* in the oil palm industry			Tween 80, a non-ionic surfactant, exhibited the lowest effective concentration (EC_50_ value of 2), indicating improved efficacy against the fungi.	[378,384]
	Phenolic Compounds	Potential as a sustainable solution for reducing *G. boninense* inoculum pressure during replantation in the oil palm industry			Benzoic acid treatment at 1 mM resulted in only 31% wood mass loss after 120 days; benzoic acid at 15 mM concentration could be a natural control agent for BSR in oil palm.	[377,391,393]
		Benzoic acid as a viable management option for BSR in oil palm production			Showed potential in suppressing *G. boninense* and preventing wood decay in oil palm trees, inhibited fungal growth, caused cellular damage, and reduced wood weight loss.	[380]
		Phenolic acid combination			A combination of phenolic acids effectively suppressed *G. boninense* and reduced *Ganoderma* colonization in palm tissues.	[398,401]
		Phenolic acid interaction			Phenolic acids interacted with *G. boninense* in oil palm roots. Ergosterol analysis showed correlation with BSR severity. Chitosan stimulated phenolic acid production.	[119]
	Herbicides	In vitro experiments			Paraquat dichloride was the most effective in inhibiting *Ganoderma* growth.	[379]
	Nanoparticles	Chitosan nanoparticles/nanoformulation offering eco-friendly control for basal stem rot			Effectively combatted *G. boninense* in oil palm trees, with no fungicide residues in palm oil; accumulated in the stem tissue and leaves, exhibiting long-lasting effects.	[381,382]
	Fungicides	Tetramethylthiuram disulfide (thiram) shows antifungal properties against *G. boninense*			Higher concentrations of thiram completely inhibited *G. boninense* growth. Ergosterol analysis confirmed the effectiveness of thiram treatment.	[383]
		Pyraclostrobin			Effectively suppressed *G. boninense*, inhibitory effects, reduced infection, and improved plant growth. Pyraclostrobin in vitro and in vivo.	[386]
		Prochloraz, kresoxim methyl, and chlorotalonil			Showed high efficacy against *G. boninense*, with percentages of mycelium growth inhibition of 96.22%, 98.96%, and 88.44	[370]
		Soil residue and dissipation of hexaconazole			Hexaconazole residues detected in soil samples, dissipating over time; half-life ranged from approximately 69.3 to 86.6 days.	[399]
		Chitosan			Effectively suppressed *Ganoderma* infection in oil palm seedlings at a minimum concentration of 0.5%, reduced disease severity, and hindered fungal sterol production.	[403]
					A 0.5% concentration reduced disease severity and fungal sterol levels. Tissue infection consistent across chitosan concentrations. Pretreatment slightly lowered severity post-infection.	[405]
	Enzyme Inhibition	Lignin-degrading enzyme inhibitors			Lignin-degrading enzymes, specifically *G. boninense* (PER71), showed high enzyme activity and effective decolorization. Enzyme inhibitors included EDTA and thioglycolic acid.	[385]
		Phenolic acids			Inhibited the ligninolytic enzyme activity of *G. boninense*. Syringic and caffeic acid showed significant inhibitory effects.	[390]
	Metabolomics	Biochemical interaction analysis			Identified asparagine and chelidonic acid as potential markers for distinguishing different treatment types in *Ganoderma* infection in oil palm seedlings.	[388]
	Nanocomposites	Zinc/aluminum-layered double hydroxide hexaconazole was successfully intercalated into zinc/aluminum-layered double hydroxide (ZALDH) using an ion-exchange method			Improved thermal stability; demonstrated enhanced inhibition against *G. boninense* growth compared to free hexaconazole.	[389]
	Plant Hormones and Mineral Nutrients	Salicylic acid	Potential as fungicides		Inhibited *G. boninense* growth.	[392]
					Concentrations of 50 and 100 ppm inhibited *Ganoderma* growth by 100%. Higher concentrations (100–200 ppm) improved seedling growth and chlorophyll content.	[408]
		Mineral nutrient and salicylic acid (SA)	Management of BSR disease		Calcium, copper, and SA treatment effectively controlled BSR disease in oil palm seedlings, with the lowest severity being 5.0%; reduced fungal activity; delayed symptoms; and promoted seedling growth.	[394]
		Mineral nutrient	Enhancement of oil palm resistance		Identified optimal N, P, and K concentrations enhanced oil palm seedlings’ resistance to *Ganoderma* disease, reducing incidence and severity in plantations.	[395]
					Cu and Ca supplementation in oil palm reduced *G. boninense* infection and promoted peroxidase and lignin production during fungal penetration.	[401]
	Plant Defense Inducers	Silicon treatment	Reduction in BSR severity		Reduced BSR severity in oil palm seedlings by 53%, showing potential economic benefits.	[396]
	Preventive Treatment	GanoCare™ OC	*Ganoderma* infection control in oil palm		Used as a preventive treatment to control *Ganoderma* infection in oil palm, reducing potential yield losses in oil palm cultivation.	[398]
		Ca^2+^, Cu^2+^, and SA	The combination demonstrates potential for suppressing *G. boninense* growth in vitro		Effectively controlled *G. boninense* in vitro. SA alone effective. Combination treatment reduced basidiocarp size, weight, and wood block loss.	[404]
	Fumigant	The fumigant methylisothiocyanate (MITC)	*Ganoderma* infection control		Restricted *Ganoderma* spread in oil palm, extending lifespan and improving productivity by eliminating inoculum spread within infected palms.	[402]
Host Resistance	Mixed Cropping	Mixed cropping with water yam	Mixed cropping shows promise as a strategy to control *G. boninense* damage.		Reduced *G. boninense* infection severity in oil palm, resulting in reduced root necrosis, lower plant mortality, and decreased pathogen inoculum potential.	[417]
	Molecular Method	Protein-coding genes and non-coding RNAs	Offering insights into infection mechanisms and disease management strategies		Protein-coding genes and non-coding RNAs play roles in *G. boninense*–oil palm interaction. Computational analysis predicted diverse non-coding RNAs.	[418]
		Marker-assisted selection (MAS)	Identification of genetic markers for resistance		Identified genetic markers for *Ganoderma* resistance in oil palm breeding, facilitating the selection of resistant planting material with reduced field infection rates.	[419]
		Genetic diversity	Insights offered for effective BSR disease control		Genetic diversity and population structure of *G. boninense* explored with cDNA-SSR markers. High variability and gene flow noted, suggesting regional adaptation and multiple genetic sources.	[197]
		Genetic engineering	Development of fungal-resistant oil palm plants		Genetic engineering created fungal-resistant oil palm plants. Two of these plants resisted *G. boninense* fungus, showing genetic engineering’s potential for disease-resistant varieties.	[421]
		Genetic loci identification	Identification of *Ganoderma* resistance loci		Genetic loci linked to *Ganoderma* resistance in oil palm identified through long-term study, aiding breeding programs in developing resistant varieties.	[34]
		Gene expression analysis	Identification of potential biomarkers		Differentially expressed genes linked to *Ganoderma* tolerance in oil palm identified, offering potential biomarkers for tolerance. Insights into genetic basis and breeding targets provided.	[425]
	Genetic Resource Discovery		Identification of *Ganoderma*-resistant materials		The Indonesian Oil Palm Research Institute has found *Ganoderma*-resistant planting materials, offering potential disease control solutions.	[422]
	Genetic Variation Study		Investigation of *G. boninense* genetic variation		Genetic variation of *G. boninense* in oil palm plantations studied, revealing a single founder population adapted to oil palm. Important for managing BSR.	[7]
	Inoculation Technique	Development of faster inoculation method	A new mycelium inoculation technique was introduced		Assessed oil palm resistance to BSR caused by *G. boninense*; allowed for consistent infection, early disease evaluation, and differentiation between resistant and susceptible palm seedlings.	[423]
	Soil Influence	Impact of soil type on disease severity			*Ganoderma* infection in oil palm influenced by soil type, affecting tree life expectancy and foliar symptoms. Improved management strategies needed, especially for palms in specific soil types.	[424]
	Breeding Program Success	Development of resistant oil palm varieties			Partial resistance to basal stem rot from *G. boninense* seen in oil palm. Breeding programs developed resistant varieties, offering a solution for managing the damaging pathogen.	[426]
	Tree Species Selection	Resistance comparison among tree species	Findings aid in selecting resistant trees		*Polyalthia longifolia* more resistant to *Ganoderma* infection than *Pterocarpus indicus*, making it a better choice for affected areas.	[427]
	Variety Susceptibility Study	Evaluation of resistance in oil palm varieties			Variability in BSR susceptibility observed in oil palm varieties. Accurate resistance assessments crucial despite assumptions about certain varieties’ resistance.	[406]
	Progeny Evaluation	Assessment of progeny resistance to BSR	Suggesting potential for developing resistant planting materials		Progenies of oil palm evaluated for *Ganoderma* resistance. Variability observed.	[429]
	Breeding Trials	Development of resistant planting material	Offering hope for disease management		Breeding trials sought *Ganoderma*-tolerant/-resistant oil palm. Genetic variation allowed for tolerant generations.	[430]
	Screening Test Development	Early selection method for resistant progenies			A screening test was developed for oil palm progenies to detect resistance to BSR. It showed reliable early selection and consistent results, and it prevented planting highly susceptible progenies, offering an efficient method for developing resistant planting materials.	[132]
	Varietal Susceptibility	Impact of varietal origin on disease susceptibility			Field observations in oil palm estates revealed varying susceptibility to BSR. Variations within origins and clones underscored the importance of early selection tests for disease management strategies.	[433]
	Progeny Resistance Assessment		Evaluation of progenies for resistance		Oil palm progenies were evaluated for resistance to *G. boninense.*	[62]
Soil Management	Soil Influence		Impact of soil type on disease severity		Phosphorus levels in BSR-infested oil palm soil affected susceptibility to *G. boninense*. Effective soil management crucial for disease control. Peat soil accelerated disease progression compared to mineral soils.	[434,435]
	Soil pH Management	Effect of soil pH on BSR disease development			pH 6 effectively suppresses BSR in oil palm seedlings. Soil pH management crucial for disease control.	[436]
	Soil Characteristics Study	Examination of soil characteristics influencing disease	Emphasizes importance of soil management in disease control		Groundwater level and pH correlated significantly with *Ganoderma* attacks in oil palm cultivation.	[332]
	Replanting Strategies		Impact of replanting methods on BSR incidence		BSR reduced with surgery-mounding and *Ganoderma* removal in oil palm replanting. Proper debris removal crucial to minimize BSR and early-stage *Ganoderma* infection.	[437]
	*Ganoderma* Survival Study		Factors influencing *Ganoderma* survival in soil		*Ganoderma* species with chlamydospores survived better at various soil moisture levels. Flooding controlled species lacking chlamydospores.	[438]
Integrated Disease Management (IDM)	Nursery Screening	Efficient methods for evaluating biopesticides	Enables efficient selection of effective biopesticides for disease management.		Nursery screening rapidly evaluates biopesticide effectiveness. Standardized parameters and experimental design crucial for accuracy.	[460]
	Disease Control Practices	Strategies for effective BSR management			Early BSR control needs calcium nitrate, fungicides, and cultural practices. Combining them is essential. Tailoring strategies optimizes results.	[442]
	Soil Amendment Study	Investigating the impact of soil amendments			Studied soil amendments like calcium nitrate on disease severity and palm health. Understanding their effect crucial for effective soil management in BSR control.	[461]

## Data Availability

Not applicable.

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
