# Peer review of "An In-Depth Study of Phytopathogenic Ganoderma: Pathogenicity, Advanced Detection Techniques, Control Strategies, and Sustainable Management"

_jof, 2024, doi:10.3390/jof10060414_

Round 1

Reviewer 1 Report

Overall, the work is well-structured, and this contribution should be considered for publication after addressing the following comments.

1.      I think the title is too long, please re-write.

2.      Graphical Abstract should be prepared to make the manuscript more attractive to Potential readers.

3.      The abstract is very short compared with the title of the article. There are no numerical values.

4.      Write a page on Taxonomy and Classification of Ganoderma species known to be phytopathogenic and also about Regulatory and Policy Considerations at the final

5.      Some graphs, figures, and tables should be added in the method analysis section

comments.

1.      I think the title is too long, please re-write.

2.      Graphical Abstract should be prepared to make the manuscript more attractive to Potential readers.

3.      The abstract is very short compared with the title of the article. There are no numerical values.

4.      Write a page on Taxonomy and Classification of Ganoderma species known to be phytopathogenic and also about Regulatory and Policy Considerations at the final

5.      Some graphs, figures, and tables should be added in the method analysis section

Author Response

Dear Reviewer,

We sincerely appreciate the time and effort you dedicated to reviewing our manuscript titled "An In-depth Study of Phytopathogenic Ganoderma: Pathogenicity, Advanced Detection Techniques, Control Strategies, and Sustainable Management." We are grateful for your insightful comments and constructive feedback, which have immensely contributed to the refinement of our work. We have made significant revisions to address each point in response to your valuable suggestions.

Please find the revised manuscript and a detailed response to each of your comments. For easy identification, we have meticulously highlighted all the corrections in blue for your convenience. We trust that these revisions have strengthened the quality and comprehensiveness of our work. Once again, we express our gratitude for your thorough review and constructive feedback. We remain committed to ensuring the highest standard of excellence in our research and look forward to your feedback on the revised manuscript.

Thank you for your continued support and consideration.

Warm regards,

Samantha Chandranath Karunarathna, Nimesha M. Patabendige, Wenhua Lu, Suhail Asad and Kalani Kanchana Hapuarachchi

Reviewer 2 Report

This is an excellent meta-analysis of the research literature surrounding Ganoderma species, the damage associated with said species, and the wide variety of diagnostic and potential management tactics. Due to the breadth of the associated research, I am a fan of detailed and comprehensive meta-analyses, which this is. This will be very valuable to both students of and researchers in this area of plant pathology.

I have attached my detailed review of the manuscript with suggested wording changes, all of which is relatively minor. You may consider removing some of the redundant sentences, but they are not overly distracting. Brevity is always good, though. 

Your concluding sections were excellent and especially the emphasis on cooerative research and management of the problem. 

I have attached the file with my detailed comments to the authors.

Author Response

Dear Reviewer,

We sincerely appreciate the time and effort you dedicated to reviewing our manuscript titled "An In-depth Study of Phytopathogenic Ganoderma: Pathogenicity, Advanced Detection Techniques, Control Strategies, and Sustainable Management." We are grateful for your insightful comments and constructive feedback, which have immensely contributed to the refinement of our work. We have made significant revisions to address each point in response to your valuable suggestions.

Please find the revised manuscript and a detailed response to each of your comments. We have meticulously highlighted all the corrections in yellow for easy identification for your convenience. We trust that these revisions have strengthened the quality and comprehensiveness of our work. Once again, we express our gratitude for your thorough review and constructive feedback. We remain committed to ensuring the highest standard of excellence in our research and look forward to your feedback on the revised manuscript.

Thank you for your continued support and consideration.

Best regards,

Samantha Chandranath Karunarathna, Nimesha M. Patabendige, Wenhua Lu, Suhail Asad and Kalani Kanchana Hapuarachchi

Reviewer 3 Report

• Unfortunately, the text lacks structure in the presentation of the material. For example, one could provide a table characterizing the distribution of known precedents of fungal pathogenesis across plant species or countries. This would give the work even more value.

• A similar point applies to pathogen control measures. It would be logical if we summarized all the applied and developed methods of combating fungi into a single table, which would also show the effectiveness of each method. In this form, the text is simply a reference to the use of one or another control method, and not a systematizing work in the field of mushroom science.

• Line 50 – what enzymes and toxins are we talking about? Specify in the text.

• There is a lack of data on genetic diversity and phylogenetic structure and relationships between different species of the genus. They need to be added to make the picture more complete and complete.

• All pictures are of poor quality and need to be improved.

• Figure 1 - due to the fact that in the title of the work, fungi of this genus are characterized as phytopathogens, then in the diagram of the life cycle of the fungus it is necessary to pay attention to the stages that the pathogen carries out on the plant.

• Figure 1 is not Ganoderma life cycle, but life cycle of Ganoderma genera fungi.

• Lines 156-158 – by what percentage does the pathogen reduce the yield and in what percentage does it cause death. In addition, there is no information on the statistics of this group of diseases in different countries.

• Section 4 – the title of the section characterizes many plants, but in fact it is devoted to only one type of plant, so it needs to be expanded or the name changed.

• Section 5 is named as pathogenicity, but the main idea is the functions of genes responsible for pathogenesis. I propose to transfer this data to the section on genetic characteristics, and to this section add a diagram of the phased destruction of the plant by the pathogen.

• Line 375 – what secondary metabolites are we talking about. Specify in the text.

• Figure 3 – if the authors took the diagram from some article, then they should indicate the link, and if they did it themselves, then improve the quality.

• Section 7 should include sections 8-23 as subsections, as this is a continuation of diagnostic methods.

• Section 17 should include a table with the primer sequences and methods reported in the section.

• Section 18 – how advisable is it to carry out spectral analysis if the pathogen that has entered the plant cannot be stopped?

• Section 24 – you must provide a table with the effectiveness of each method and links to the works in which they are described.

• Section 28 – how the authors are going to stop the pathogen if it is inside the tree trunk, and biological products act only by contact and superficially.

• Section 31 – does the pathogen have a racial structure, according to a gene-by-gene pattern? Correct it in the text.

Author Response

Dear Reviewer,

We sincerely appreciate the time and effort you dedicated to reviewing our manuscript titled "An In-depth Study of Phytopathogenic Ganoderma: Pathogenicity, Advanced Detection Techniques, Control Strategies, and Sustainable Management." We are grateful for your insightful comments and constructive feedback, which have immensely contributed to the refinement of our work. We have made significant revisions to address each point in response to your valuable suggestions.

Please find the revised manuscript and a detailed response to each of your comments. For easy identification, we have meticulously highlighted all the corrections in green for your convenience. We trust that these revisions have strengthened the quality and comprehensiveness of our work. Once again, we express our gratitude for your thorough review and constructive feedback. We remain committed to ensuring the highest standard of excellence in our research and look forward to your feedback on the revised manuscript.

Thank you for your continued support and consideration.

Best Regards,

Samantha Chandranath Karunarathna, Nimesha M. Patabendige, Wenhua Lu, Suhail Asad and Kalani Kanchana Hapuarachchi
